# Genome-wide detection of imprinted differentially methylated regions using nanopore sequencing

Vahid Akbari[1,2], Jean-Michel Garant[1], Kieran O'Neill[1], Pawan Pandoh[1], Richard Moore[1], Marco A Marra[1,2], Martin Hirst[1,3], Steven JM Jones[1,2]*

[1]Canada's Michael Smith Genome Sciences Centre, BC Cancer Agency, Vancouver, Canada; [2]Department of Medical Genetics, University of British Columbia, Vancouver, Canada; [3]Department of Microbiology and Immunology, Michael Smith Laboratories, University of British Columbia, Vancouver, Canada

**Abstract** Imprinting is a critical part of normal embryonic development in mammals, controlled by defined parent-of-origin (PofO) differentially methylated regions (DMRs) known as imprinting control regions. Direct nanopore sequencing of DNA provides a means to detect allelic methylation and to overcome the drawbacks of methylation array and short-read technologies. Here, we used publicly available nanopore sequencing data for 12 standard B-lymphocyte cell lines to acquire the genome-wide mapping of imprinted intervals in humans. Using the sequencing data, we were able to phase 95% of the human methylome and detect 94% of the previously well-characterized, imprinted DMRs. In addition, we found 42 novel imprinted DMRs (16 germline and 26 somatic), which were confirmed using whole-genome bisulfite sequencing (WGBS) data. Analysis of WGBS data in mouse (*Mus musculus*), rhesus monkey (*Macaca mulatta*), and chimpanzee (*Pan troglodytes*) suggested that 17 of these imprinted DMRs are conserved. Some of the novel imprinted intervals are within or close to imprinted genes without a known DMR. We also detected subtle parental methylation bias, spanning several kilobases at seven known imprinted clusters. At these blocks, hypermethylation occurs at the gene body of expressed allele(s) with mutually exclusive H3K36me3 and H3K27me3 allelic histone marks. These results expand upon our current knowledge of imprinting and the potential of nanopore sequencing to identify imprinting regions using only parent-offspring trios, as opposed to the large multi-generational pedigrees that have previously been required.

*For correspondence:
sjones@bcgsc.ca

**Competing interest:** The authors declare that no competing interests exist.

## Editor's evaluation

This work uses nanowire sequencing to detect genome-Wide imprinted differentially methylated regions. It will be of broad interest to DNA methylation researchers.

## Introduction

The addition of a methyl group to the fifth carbon of cytidine is the most prevalent and stable epigenetic modification of human DNA (*Laurent et al., 2010*). DNA methylation is involved in gene regulation and influences a vast array of biological mechanisms, including embryonic development and cell fate, genome imprinting, X-chromosome inactivation, and transposon silencing (*Moore et al., 2013*; *Smith and Meissner, 2013*). In mammals, there are two copies or alleles of a gene, one inherited from each parent. Most gene transcripts are expressed from both alleles. However, a subset of genes are only expressed from a single allele; this allele can be selected either randomly, as seen

in X-chromosome inactivation in females, or based upon the parent-of-origin (PofO), referred to as imprinting (*Chess, 2013*; *Khamlichi and Feil, 2018*).

In imprinting, mono-allelic expression of a gene or cluster of genes is controlled by a cis-acting imprinting control region (ICR) (*Bartolomei and Ferguson-Smith, 2011*). The main mechanism by which this occurs is PofO-defined differential methylation at ICRs, also known as imprinted differentially methylated regions (DMRs) (*Bartolomei and Ferguson-Smith, 2011*; *Maupetit-Méhouas et al., 2016*). Imprinted DMRs are classified as germline (primary) or somatic (secondary), hereinafter referred to as gDMR and sDMR. gDMRs are established during the first wave of methylation reprogramming in germ cell development and escape the second methylation reprogramming after fertilization (*Zink et al., 2018*). sDMRs are established de novo after fertilization during somatic development, usually under the control of a nearby gDMR (*Zink et al., 2018*). Imprinted clusters of genes may span up to ~4 Mb, by acting through a CCCTC-binding factor (CTCF)-binding site or by allelic expression of a long non-coding RNA (*Bartolomei and Ferguson-Smith, 2011*; *da Rocha and Gendrel, 2019*). By contrast, individually imprinted genes are typically regulated by PofO-derived differential methylation at the gene promoter (*Bartolomei and Ferguson-Smith, 2011*).

Imprinting is implicated in various genetic disorders, either from aberrations in imprinted methylation or from deleterious variants affecting the ICR and imprinted genes. Aberrant imprinted methylation is also detected in several human cancers (*Goovaerts et al., 2018*; *Jelinic and Shaw, 2007*; *Tomizawa and Sasaki, 2012*). Thus, accurate mapping and characterization of imprinting in humans is key to the treatment and actionability of genetic disorders, and to personalized oncogenomics.

To detect imprinted methylation, accurate assignment of methylation data to paternal and maternal alleles is required. Achieving this with traditional bisulfite sequencing or arrays is challenging. Several studies have used samples with large karyotypic abnormalities, such as uniparental disomies, teratomas, and hydatidiform moles, to infer regions of imprinting (*Court et al., 2014*; *Hernandez Mora et al., 2018*; *Joshi et al., 2016*). This approach relies not only on rare structural variants, but also on the assumption that both normal methylation and the imprinted state remain intact in spite of substantial genomic aberrations. A study by *Zink et al., 2018*, leveraged a genotyped, multi-generation pedigree spanning nearly half the population of Iceland (n=150,000), in combination with whole-genome oxidative bisulfite sequencing, to phase methylation and infer PofO (*Zink et al., 2018*). However, despite being able to phase nearly every SNP in that cohort, they were only able to phase 84% of the human autosomal methylome in over 200 samples due to the short length of reads. Furthermore, the study was based on a single, genetically isolated population, which may not be representative of the wider human population. A comprehensive mapping of imprinted methylation using a technology more suited to phasing reads, based on individuals more representative of the human population, could greatly advance our understanding of imprinting, with direct benefits for human health.

Previously, we have shown that nanopore sequencing can detect allelic methylation in a single sample and accurately determine PofO using only trio data. We also previously developed the software NanoMethPhase for this purpose (*Akbari et al., 2021*). Here, we applied NanoMethPhase to public nanopore data from a diverse set of 12 lymphoblastoid cell lines (LCLs) from the 1000 Genomes Project (1KGP) and Genome in a Bottle (GIAB) to investigate genome-wide allele-specific methylation (ASM) and detect novel imprinted DMRs (*Figure 1A*; *Auton et al., 2015*; *De Coster et al., 2019*; *Jain et al., 2018*; *Shafin et al., 2020*; *Zook et al., 2016*; *Zook et al., 2019*). Using trio data from GIAB and 1KGP for these cell lines, we phased nanopore long reads to their PofO and inferred allelic methylation (*Akbari et al., 2021*; *Auton et al., 2015*; *Zook et al., 2019*). We were able to detect haplotype and methylation status for 26.5 million autosomal CpGs comprising 95% of the human autosomal methylome (GRCh38 main chromosomes). We further used public whole-genome bisulfite sequencing (WGBS) data to confirm the presence of the detected DMRs in other tissues and to classify the novel DMRs as germline or somatic. We captured 94% of the well-characterized DMRs and detected 42 novel DMRs (16 gDMRs and 26 sDMRs). Of these novel DMRs, 40.5% show evidence of conservation. We also detected seven blocks of PofO methylation bias at seven imprinted clusters with mutual exclusive allelic H3K36me3 and H3K27me3 histone marks. Collectively, our results extend the set of known imprinted intervals in humans and demonstrate a major contribution in our ability to characterize imprinting by ASM, brought about by the capabilities of long-read nanopore sequencing.

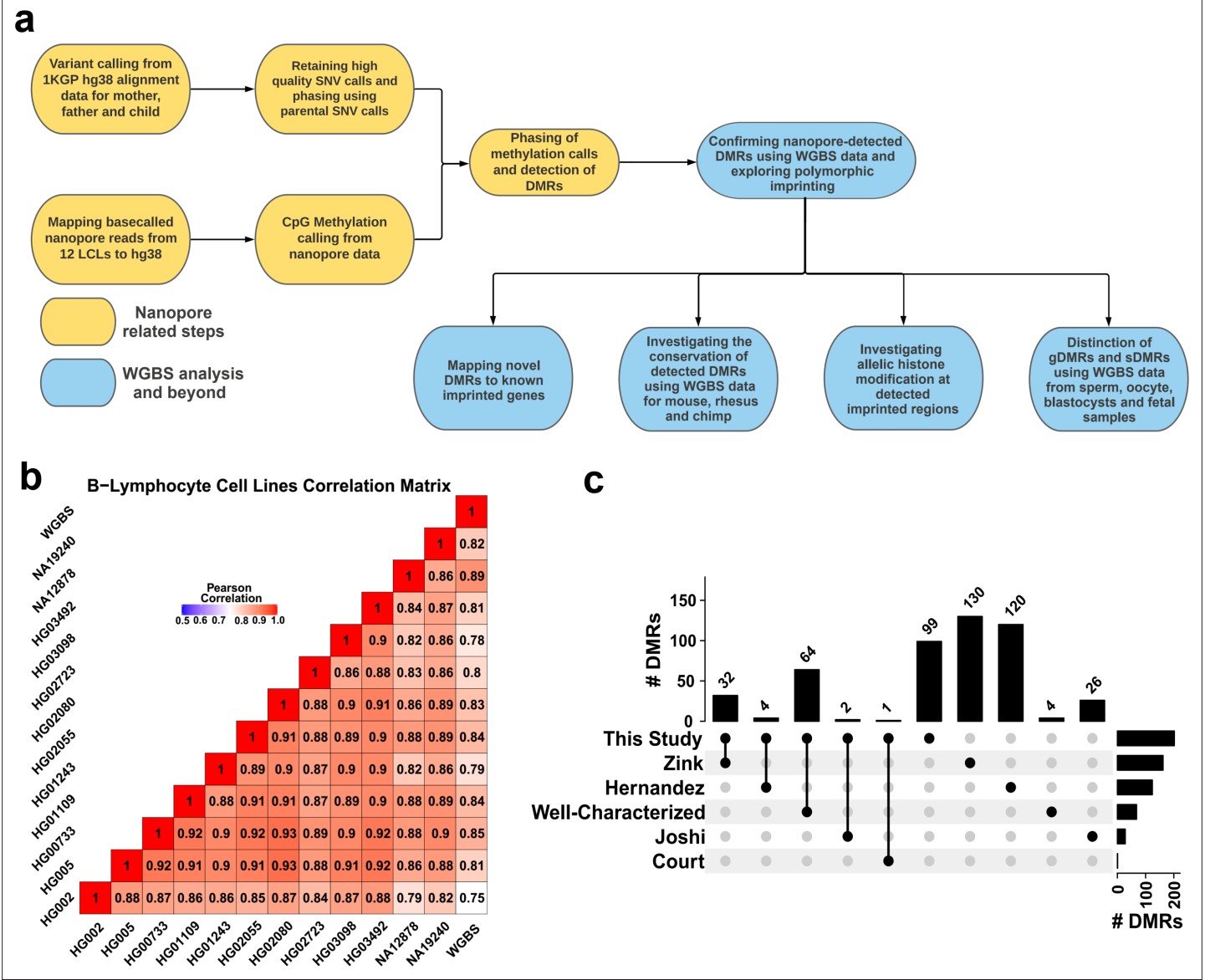

**Figure 1.** Detection of allelic methylation using nanopore sequencing. (**a**) Flowchart of the study representing all the analysis steps. (**b**) Pearson correlation matrix of the nanopore CpG methylation frequencies for the 12 cell lines and NA12878 whole-genome bisulfite sequencing (WGBS) from ENCODE (ENCFF835NTC). (**c**) Upset plot of the number of differentially methylated regions (DMRs) detected in our study and previous studies, including overlaps.

The online version of this article includes the following figure supplement(s) for figure 1:

**Figure supplement 1.** Number of allelic differentially methylated regions (DMRs) overlapped to the reported DMRs and parent-of-origin (PofO)-defined phased CpG methylation in each sample examined for differential methylation analysis by DSS R package.

## Results

### Assessing the effectiveness of nanopore methylation calling and detection of known imprinted DMRs

Using the set of 12 LCLs for which we called methylation data, we conducted correlation analysis among nanopore-called methylation data and another WGBS dataset for NA12878 cell line (ENCFF835NTC) to confirm the reliability of methylation calling (*Figure 1B*). We observed high correlation across cell lines (r=0.75–0.93), as expected as they were the same cell type. NA12878 nanopore-called methylation also showed the highest correlation (r=0.89) with NA12878 WGBS, as expected (*Figure 1B*). To assess the use of nanopore long reads in detecting known DMRs, we identified previously reported

DMRs, including 383 imprinted intervals (*Court et al., 2014*; *Hernandez Mora et al., 2018*; *Joshi et al., 2016*; *Zink et al., 2018*). Of these 383, we classified 68 as 'well-characterized' as they were reported by at least two genome-wide mapping studies or were previously known to be imprinted (*Supplementary file 1*). Subsequently, we haplotyped the methylome in each cell line, and performed differential methylation analysis (DMA) between alleles across cell lines; 95% (26.5M) of the human autosomal CpGs could be assigned to a haplotype. We detected 200 allelic DMRs (p-value <0.001, |methylation difference|>0.20, and detected in at least four cell lines in each haplotype) (*Supplementary file 2*). Out of the 200 detected DMRs, 101 overlapped with 103 previously reported DMRs with consistent PofO (*Supplementary file 3*), while the remaining 99 were novel (*Figure 1C*). Of the well-characterized DMRs, 64/68 (94%) were detected in our study (*Figure 1C*; *Supplementary file 3*).

Similarly, we assessed methylation haplotyping and detection of imprinted DMRs within a single sample. On average, 90% (M ± SD = 25 M ± 1.61 M) of the human methylome could be assigned to a parental haplotype in each cell line (*Figure 1—figure supplement 1*). Among the well-characterized DMRs, ~73% (M ± SD = 49.5 ± 4.5) could be detected in a single cell line. An additional 33 DMRs (SD = 9.6) reported by only one previous study were detected in each cell line (*Figure 1—figure supplement 1*).

## Confirmation of novel imprinted DMRs

We detected 99 imprinted DMRs that did not overlap with previously reported imprinted DMRs (*Court et al., 2014*; *Hernandez Mora et al., 2018*; *Joshi et al., 2016*; *Zink et al., 2018*). In order to confirm these DMRs in human tissues and detect potential novel imprinted regions, we investigated WGBS datasets for partial methylation at nanopore-detected DMRs (Materials and methods). We used 60 WGBS datasets from 29 tissue types and 119 blood samples from 87 individuals (*Bernstein et al., 2010*; *ENCODE Project Consortium, 2012*; *Stunnenberg et al., 2016*). We first examined the 68 well-characterized DMRs, 91% of them demonstrated partial methylation (more than 60% of the CpGs at the DMR having between 0.35 and 0.65 methylation) in at least one tissue and individual blood samples (*Figure 2A and B*). As controls, we used 100 randomly selected 1, 2, 3 kb bins, and CpG islands (CGIs) in 100 resampling iterations. Of these, 0.65%, 0.74%, 2.28%, and 4.83% of the randomly selected 3, 2, 1 kb, and CGIs, respectively, demonstrated partial methylation (*Figure 2—figure supplement 1*). Applying this approach to the 99 previously unreported DMRs, the WGBS data supported 42 of the novel DMRs (*Figure 2*, *Table 1*). In agreement with previous studies reporting a higher number of maternally methylated intervals (*Court et al., 2014*; *Hernandez Mora et al., 2018*; *Joshi et al., 2016*), 74% of the novel DMRs were maternally methylated. Overall, we detected 143 imprinted DMRs of which 101 were found to overlap with previously reported DMRs while 42 were novel DMRs detected by nanopore and confirmed using WGBS data (*Figure 2C*, *Supplementary file 4*).

## Novel imprinted DMRs display inter-individual variation

Although imprinted methylation is generally regarded as consistent between individuals and resistant to environmental factors, there are examples of polymorphic imprinting where imprinted methylation is not consistently observed across individuals. In order to assess the inter-individual variation of the novel imprinted DMRs, we examined partial methylation in the 119 blood samples from 87 individuals. Some imprinted DMRs such as *VTRNA2-1*, *IGF2*, *RB1*, *PARD6G*, *CHRNE*, and *IGF2R* are known to be polymorphic (*Joshi et al., 2016*; *Zink et al., 2018*). The detected DMRs that mapped to these imprinted regions displayed partial methylation in 2–65% of the individuals in our analysis (M ± SD = 40% ± 22%; *Supplementary file 5*). *ZNF331* DMR is known to be consistently imprinted across individuals (*Zink et al., 2018*). In our analysis, the DMR that mapped to *ZNF331* reported interval displayed partial methylation in 99% of the individuals (*Supplementary file 5*). We then examined inter-individual variation across the 42 novel DMRs. Imprinted methylation at all the novel DMRs demonstrated variation ranging from 1.2% to 73.5% of the individuals (M ± SD = 23.6% ± 19.2%; *Table 1*). Among the novel DMRs, maternal sDMR near *BTBD7P1* is the most consistent with partial methylation in 73.5% of the individuals (*Table 1*). On the other hand, the novel paternal sDMRs within *AC092296.3* and *UBAC2* are the most variable with partial methylation in 1.2% of the individuals (*Table 1*). Among the individuals, four displayed hypermethylation at several of the well-characterized

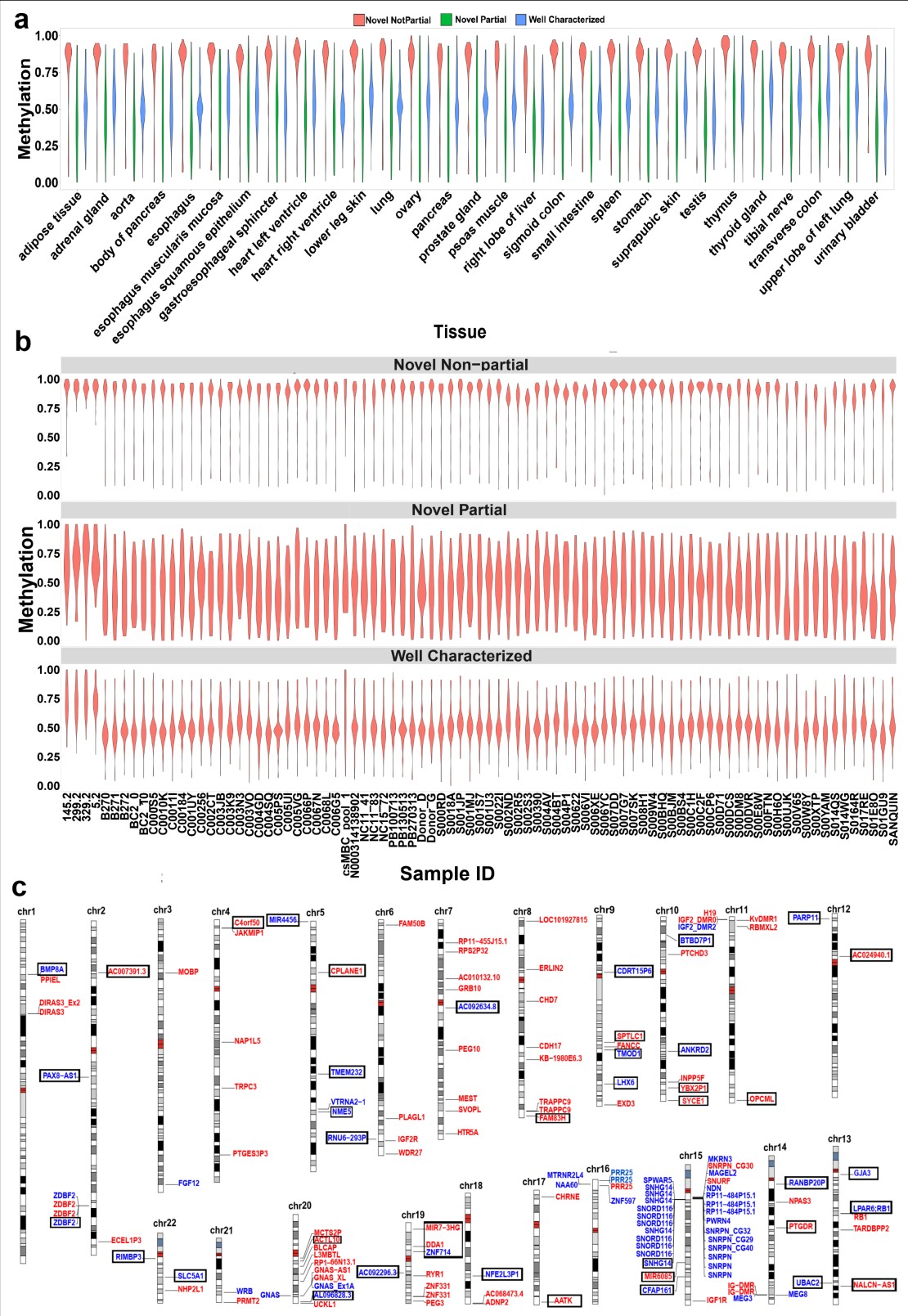

**Figure 2.** Confirmation of nanopore-detected differentially methylated regions (DMRs) using whole-genome bisulfite sequencing (WGBS) data. (**a**) and (**b**) Violin plots representing the average methylation of each DMR in WGBS tissue and blood samples. (**c**) Idiogram of the 101 DMRs overlapping to reported intervals and 42 novel DMRs which were confirmed by WGBS. Paternally methylated DMRs are labelled on the left side of each chromosome

*Figure 2 continued on next page*

*Figure 2 continued*

while maternally methylated DMRs are on the right. Red labels represent germline DMRs while blue labels represent somatic DMRs. Novel DMRs are boxed and named based on their nearest gene (Ensembl Gene 103 GRCh38.p13).

The online version of this article includes the following source data and figure supplement(s) for figure 2:

**Source data 1.** The average methylation of each DMR in human WGBS tissue samples.

**Source data 2.** The average methylation of each DMR in human WGBS blood samples.

**Figure supplement 1.** Violin plots representing methylation in whole-genome bisulfite sequencing (WGBS) blood (left) and tissue (right) samples at randomly selected CpG islands (CGIs), 1, 2, and 3 kb intervals.

and novel DMRs (*Figure 2B*), in line with a previous study that identified rare individuals with consistent hyper- or hypomethylation at dozens of imprinted loci, indicative of a generalized imprinting disruption (*Joshi et al., 2016*).

As demonstrated in *Figure 1C*, a considerable number of imprinted DMRs detected in different studies are not overlapping between studies. Different studies used different samples and individuals, therefore, we examined inter-individual variation at DMRs detected in two or more studies (including the current work) and those that detected in one study (*Supplementary file 5*). The DMRs that detected in at least two studies demonstrated more consistency across individuals (M ± SD = 41.2% ± 33%) while DMRs detected in a single study showed more variability (M ± SD = 10.6% ± 15.4%) (*Supplementary file 5*). These results suggest that polymorphic imprinting can explain this non-overlapping DMRs across studies.

## Determination of germline versus somatic status of novel imprinted DMRs

We investigated the methylation status of the detected novel DMRs in sperm and oocyte to determine if they are germline or somatic imprinted intervals. Maternally methylated gDMRs must display high methylation in oocyte and very low or no methylation in sperm with partial methylated after fertilization. Paternally methylated gDMRs must show high methylation in sperm and very low or no methylation in oocyte with partial methylated after fertilization. For the novel DMRs, 16 were detected as germline (more than 70% methylation in oocyte and less than 20% in sperm and vice versa) from which 15 were maternally methylated and one was paternally methylated (*Figure 3A and B*). Moreover, novel candidate gDMRs showed partial methylation in the blastocyst and fetal samples, indicating the gDMRs escaped de-methylation after fertilization. Meanwhile, sDMRs displayed partial methylation in fetal tissues, indicating their establishment during somatic development (*Figure 3A and B*). Overall, 16 of the novel DMRs were found to be germline while 26 were sDMRs.

During germ cell development, gDMRs are bound by proteins critical for their methylation maintenance during post-fertilization reprogramming. *ZFP57* and *ZNF445* have been identified as imprinting maintenance proteins (*Takahashi et al., 2019*). Using *ZFP57* and *ZNF445* ChIP-seq peak calling information from human embryonic stem cells and the HEK 293T cell line (*Imbeault et al., 2017*; *Takahashi et al., 2019*), 44% of the novel gDMRs and 49% of the reported gDMRs were bound by *ZFP57* and/or *ZNF445* (*Figure 3C*; *Supplementary file 4*). Of these gDMRs, 89% had a *ZFP57* peak and 45% had a *ZNF445* peak. This highlights the importance of *ZFP57* as an important factor for the maintenance of imprinted methylation at gDMRs. 5'-TGC(5mC)GC-3' is the canonical binding motif for *ZFP57* (*Quenneville et al., 2011*). Eighty-eight percent of the gDMRs with a *ZFP57* peak had at least one 5'-TGCCGC-3' motif, while 40% of the gDMRs without *ZFP57* peak had at least one 5'-TGCCGC-3' motif in the human genome (GRCh38; *Supplementary file 4*). Moreover, at gDMRs the number of 5'-TGCCGC-3' motifs demonstrated a significant positive correlation with the number of individuals demonstrating partial methylation (Pearson = 0.54, p-value = 3.6e−07; *Appendix 1—figure 1*). This suggests that a greater number of motifs provide more functional binding opportunities for *ZFP57* and also less likelihood that all *ZFP57* motifs could be perturbed through polymorphism or DNA sequence variation resulting in the imprinted methylation being less polymorphic.

## Allelic H3K4me3 histone mark at detected DMRs

The H3K4me3 histone mark is protective against DNA methylation. At imprinted DMRs, the unmethylated allele is usually enriched for this histone modification (*Court et al., 2014*; *John and Lefebvre,*

**Table 1.** Forty-two detected imprinted differentially methylated regions (DMRs) from nanopore data and confirmed using whole-genome bisulfite sequencing (WGBS) data.
DMRs are named after the nearest gene (EnsemblGene 103 GRCh38.p13).

| ID | DMR name | Origin | Type | Distance to nearest imprinted gene | % Individuals with partial methylation | % Tissues with partial methylation |
|----|----------|--------|------|------|------|------|
| 22 | AC024940.1 | Maternal | Germline | 0 | 15.9 | 3.8 |
| 35 | DDA1 | Maternal | Germline | 0 | 7.3 | 15.4 |
| 38 | ACTL10;NECAB3 | Maternal | Germline | 0 | 3.7 | 8.7 |
| 42 | SYCE1 | Maternal | Germline | 3.2 kb | 4 | 9.1 |
| 12 | FAM83H | Maternal | Germline | 149 kb | 48.8 | 12 |
| 20 | OPCML | Maternal | Germline | 744.1 kb | 45.1 | 25 |
| 19 | YBX2P1 | Maternal | Germline | >2 Mb | 3.7 | 7.7 |
| 26 | NALCN-AS1 | Maternal | Germline | >2 Mb | 30.5 | 10 |
| 28 | PTGDR | Maternal | Germline | >2 Mb | 8.4 | 3.4 |
| 32 | AATK | Maternal | Germline | >2 Mb | 23.2 | 9.1 |
| 34 | MIR7-3HG | Maternal | Germline | >2 Mb | 8.1 | 3.6 |
| 2 | AC007391.3 | Maternal | Germline | >2 Mb | 37.2 | 60.7 |
| 5 | C4orf50 | Maternal | Germline | >2 Mb | 14.5 | 22.2 |
| 7 | CPLANE1 | Maternal | Germline | >2 Mb | 2.3 | 7.1 |
| 14 | SPTLC1 | Maternal | Germline | >2 Mb | 5.8 | 48.1 |
| 1 | BMP8A | Maternal | Somatic | 0 | 4.5 | 26.1 |
| 24 | LPAR6;RB1 | Maternal | Somatic | 0 | 2.3 | 10.3 |
| 36 | ZNF714 | Maternal | Somatic | 0 | 43.9 | 29.6 |
| 17 | BTBD7P1 | Maternal | Somatic | >2 Mb | 73.5 | 55.6 |
| 18 | ANKRD2 | Maternal | Somatic | >2 Mb | 34.1 | 3.8 |
| 23 | GJA3 | Maternal | Somatic | >2 Mb | 25.6 | 21.4 |
| 27 | RANBP20P | Maternal | Somatic | >2 Mb | 28 | 32 |
| 33 | NFE2L3P1 | Maternal | Somatic | >2 Mb | 29.3 | 44.4 |
| 39 | AL096828.3 | Maternal | Somatic | >2 Mb | 50 | 7.7 |
| 41 | SLC5A1 | Maternal | Somatic | >2 Mb | 56.1 | 25.9 |
| 8 | TMEM232 | Maternal | Somatic | >2 Mb | 25.6 | 37.9 |
| 9 | NME5 | Maternal | Somatic | >2 Mb | 22.1 | 10.7 |
| 11 | AC092634.8 | Maternal | Somatic | >2 Mb | 6.1 | 3.8 |
| 13 | CDRT15P6 | Maternal | Somatic | >2 Mb | 12.7 | 5.6 |
| 15 | TMOD1 | Maternal | Somatic | >2 Mb | 35.4 | 14.8 |
| 16 | LHX6 | Maternal | Somatic | >2 Mb | 44.6 | 25 |
| 30 | MIR6085 | Paternal | Germline | >2 Mb | 27.1 | 25.9 |
| 3 | PAX8;PAX8-AS1 | Paternal | Somatic | 0 | 24.4 | 32.1 |
| 4 | ZDBF2 | Paternal | Somatic | 0 | 53.6 | 58.6 |
| 29 | SNHG14 | Paternal | Somatic | 3 kb | 4.7 | 37.9 |
| 37 | AC092296.3 | Paternal | Somatic | 90 kb | 1.2 | 7.7 |

*Table 1 continued on next page*

*Table 1 continued*

| ID | DMR name | Origin | Type | Distance to nearest imprinted gene | % Individuals with partial methylation | % Tissues with partial methylation |
|----|----------|--------|------|-------|------|------|
| 40 | RIMBP3 | Paternal | Somatic | 296 kb | 17.4 | 11.1 |
| 10 | RNU6-293P | Paternal | Somatic | 1.03 Mb | 65.1 | 37 |
| 21 | PARP11 | Paternal | Somatic | >2 Mb | 10.8 | 22.2 |
| 25 | UBAC2 | Paternal | Somatic | >2 Mb | 1.2 | 6.9 |
| 31 | CFAP161 | Paternal | Somatic | >2 Mb | 22.4 | 6.9 |
| 6 | MIR4456 | Paternal | Somatic | >2 Mb | 11.3 | 22.2 |

*2011*). We used H3K4me3 chromatin immunoprecipitation sequencing (ChIP-seq) data for six LCLs and their heterozygous single-nucleotide variant (SNV) calls from 1KGP. Fifty of the DMRs mapped to reported intervals and 19 of the novel DMRs could be examined. Of these, 47 previously reported and 16 novel DMRs showed a significant allelic count in ChIP-seq data (Fisher's combined p-value binomial <0.05 with at least 80% of the reads on one allele) (*Figure 4a*; *Supplementary file 6*). We also examined if the allelic H3K4me3 and methylation are in opposite alleles in NA12878 and NA19240. Forty of the previously reported DMRs and 10 of the novel DMRs with significant allelic H3K4me3 could be examined in NA12878 and/or NA19240. Thirty-seven previously reported and seven novel DMRs showed opposite allelic states between H3K4me3 and methylation (*Figure 4b*; *Supplementary file 6*).

Overall, gDMRs were enriched more with the H3K4me3 mark. Sixty-three percent of the gDMRs and 48% of the sDMRs with at least one heterozygous SNV demonstrated an allelic H3K4me3 mark (*Supplementary file 4*). This is consistent with previous studies demonstrating the protective role of H3K4me3 against DNA methylation, specifically at germline ICRs in the second round of re-methylation during implantation and somatic development (*Chen and Zhang, 2020*; *Hanna and Kelsey, 2014*).

## Conservation of detected imprinted DMRs across mammals

To investigate the conservation of detected DMRs and determine if any of the novel DMRs are conserved in mammals, we used WGBS data from mouse (*Mus musculus*), rhesus macaque (*Macaca mulatta*), and chimpanzee (*Pan troglodytes*) (*Hon et al., 2013*; *Jeong et al., 2021*; *Tung et al., 2012*). In determining whether any of the orthologous regions in these mammals displayed partial methylation, we found that 81 of the detected intervals which overlapped with previously reported DMRs and 17 of the novel imprinted DMRs displayed partial methylation in at least one tissue sample in one or more mammals (*Figure 5A*; *Supplementary file 4*). In the mouse, orthologs of the 33 detected DMRs were partially methylated, 20 of these were previously reported to be imprinted in mice (*Gigante et al., 2019*; *Xie et al., 2012*). Most (88%) of the partially methylated DMRs in the mouse were also partially methylated in rhesus macaque and/or chimpanzee suggesting conservation across species. These shared DMRs mapped to well-known imprinted clusters including *KCNQ1, H19, GNAS*, SNURF/*SNRPN, PLAGL, SGCE, BLCAP, PEG3, PEG10, PEG13, GRB10, BLCAP, NAP1L5, INPP5F,* and *MEG3* where their allelic PofO expression has already been reported in mouse and other mammals (*Geneimprint, 2021*; *Morison et al., 2001*).

Sperm, oocyte, and embryo WGBS data for mouse and rhesus macaque were used to investigate if DMRs classified as germline or somatic in humans were still germline or somatic in other mammals and vice versa (*Figure 5B*; *Dahlet et al., 2020*; *Gao et al., 2017*; *Jung et al., 2017*; *Saenz-de-Juano et al., 2019*). Overall, imprinted DMRs preserved their identity as germline or somatic in the two other mammals examined (*Figure 5B*). However, in a few cases, the type of imprinted DMR was not consistent between humans and other mammals (*Figure 5B*). This finding is supported by an earlier study indicated that imprinting is largely conserved in mammals while the identity of ICR at the germline stage is not completely conserved (*Cheong et al., 2015*).

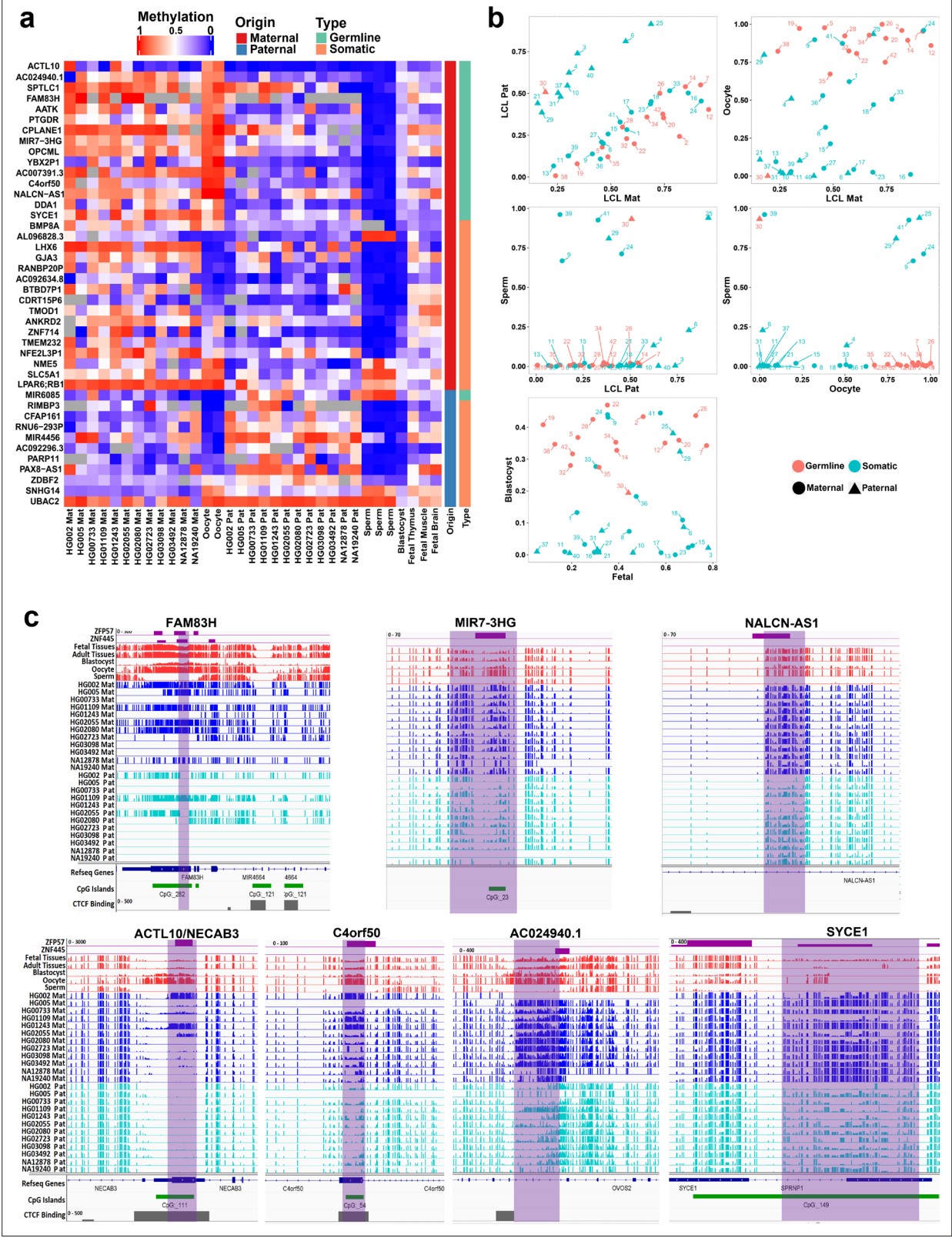

**Figure 3.** Detection of novel germline and somatic differentially methylated regions (DMRs). (**a**) Heatmap displaying average methylation of the 42 nanopore-detected DMRs confirmed by whole-genome bisulfite sequencing (WGBS). DMRs are named based on their nearest gene (Ensembl Gene 103 GRCh38.p13). (**b**) Dot plots representing the methylation of novel germline and somatic DMRs in each sample with respect to other samples. (**c**) IGV screenshots from six novel germline DMRs overlapping with *ZNF445* and/or *ZFP57* chromatin immunoprecipitation sequencing (ChIP-seq) peaks. The range for all methylation tracks is 0–1.

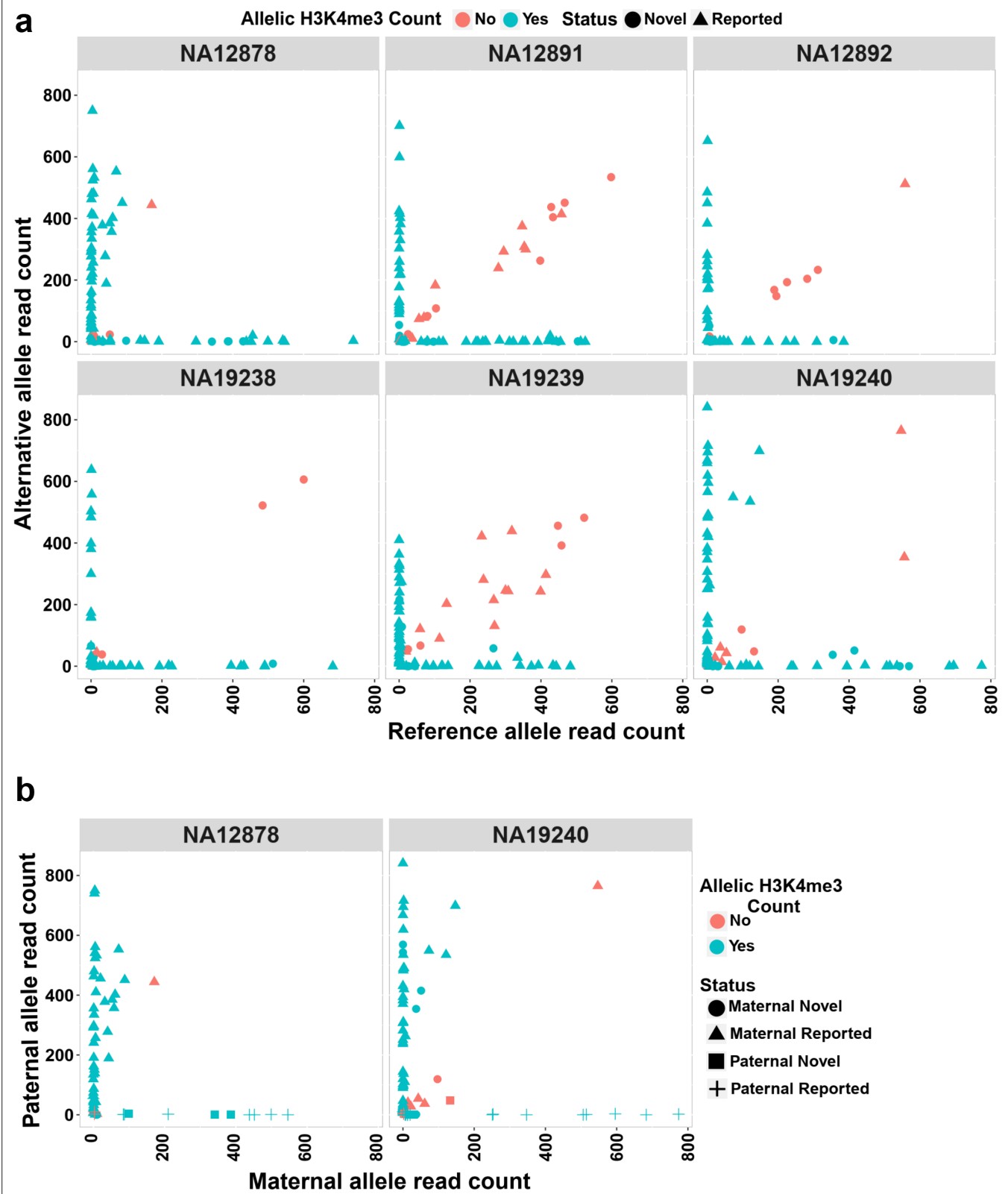

**Figure 4.** Allelic H3K4me3 histone mark at detected imprinted differentially methylated regions (DMRs). (**a**) The plots representing reference and alternative alleles H3K4me3 read counts for the heterozygous single-nucleotide variants (SNVs) mapped to the detected DMRs for the six examined samples. Each point represents an SNV. Blue color displays SNVs with Fisher's combined p-value binomial <0.05 and at least 80% of the reads on one allele and red color represent those SNVs that did not satisfy either of these thresholds. (**b**) The plots representing paternal and maternal H3K4me3 read

*Figure 4 continued on next page*

*Figure 4 continued*

counts for the heterozygous SNVs at DMRs in NA12878 and NA19240. Each point represents an SNV. The 'Status' indicates the methylation origin of the DMR and if the DMR is novel or reported.

The online version of this article includes the following source data for figure 4:

**Source data 1.** H3K4me3 allelic read counts for the heterozygous single-nucleotide variants (SNVs) mapped to the detected DMRs.

**Source data 2.** H3K4me3 allelic read counts for the paternal and maternal heterozygous single-nucleotide variants (SNVs) mapped to the detected DMRs.

## Novel DMRs within known imprinted gene domains

To examine the vicinity of novel DMRs to known imprinted genes, we assembled a list of 259 imprinted genes identified in previous studies (*Supplementary file 7*; *Babak et al., 2015*; *Baran et al., 2015*; *Geneimprint, 2021*; *Jadhav et al., 2019*; *Morison et al., 2001*; *Zink et al., 2018*). Fifteen of the novel DMRs (six germline and nine somatic) identified in our study could be mapped nearby (<1.03 Mb) to known imprinted genes (*Table 1*; *Supplementary file 4*).

Novel sDMRs close to known imprinted genes were mostly paternal of origin. Five of them mapped within known imprinted genes including *ZDBF2*, *PAX8/PAX8-AS1*, *LPAR6/RB1*, *BMP8A*, and *ZNF714* while four mapped close to imprinted genes including *PWAR1*, *LINC00665*, *DGCR6*, and *IGF2R* (*Figure 6*; *Figure 6—figure supplements 1–7*). For *ZNF714* and *PAX8/PAX8-AS1*, there are no reported imprinted DMRs within the gene or very close to them that explain their imprinted expression. Two of the novel sDMRs mapped to the promoters of these genes with a reverse relation between origin of methylation and expression (*Figure 6*), suggesting these DMRs could directly suppress paternal and maternal alleles in *PAX8-AS1* and *ZNF714*, respectively.

All novel gDMRs close to imprinted genes were maternal of origin. Three of them mapped within known imprinted genes including *ACTL10/NECAB3*, *DDA1*, and *AC024940.1* while three of them mapped close to imprinted genes including *SYCE1*, *NAPRT*, and *NTM* (*Figure 7*; *Figure 7—figure supplements 1–4*). Three of the germline DMRs mapped within or very close to three known imprinted genes without reported ICR including *AC024940.1* (*OVOS2*), *ACTL10/NECAB3,* and *SYCE1*. A novel maternal gDMR mapped to the promoter of the paternally expressed *ACTL10* (*Zink et al., 2018*; *Figure 7A*). In a previous study, a CpG site located ~130 bp away from the DMR we detected was demonstrated to be a cis-methylation quantitative trait loci with PofO association (*Cuellar Partida et al., 2018*). Thus, the novel gDMR might be the ICR of this gene and directly suppress the maternal allele. Another novel maternal gDMR mapped to the promoter of *SYCE1*, which demonstrates paternal expression bias in the allele-specific expression (ASE) track (*Zink et al., 2018*; *Figure 7B*). *Nakabayashi et al., 2011*, also observed two array probes consistent with an imprinted DMR at this region, but were unable to validate them because of the difficulty in designing bisulfite PCR primers (*Nakabayashi et al., 2011*). The novel maternal gDMR at the promoter of *SYCE1* could be the ICR for this gene and directly suppress the maternal allele.

## Contiguous blocks of parental methylation bias

Previous studies demonstrated two paradigms of imprinting at the PWS/AS imprinted cluster, either PofO methylation confined to particular regulatory regions such as CGIs or subtle paternal bias across this cluster with spikes of maternal methylation (*Court et al., 2014*; *Joshi et al., 2016*; *Sharp et al., 2010*; *Zink et al., 2018*). Probes with paternal methylation bias at the *SNORD116* cluster have been reported, spanning about a 95 kb region, and paternal deletion of this cluster results in PWS phenotypes (*Hernandez Mora et al., 2018*; *Joshi et al., 2016*; *Matsubara et al., 2019*). Slight hypomethylation of *SNORD116* cluster in cases with PWS phenotype and hypermethylation in the cases with AS phenotype have been reported (*Matsubara et al., 2019*). We did not observe paternal methylation bias across the whole PWS/AS cluster; however, we detected a paternal methylation block spanning ~200 kb, immediately downstream of the known, maternally methylated PWS *SNURF/SNRPN* ICR (*Figure 8*). This block encompasses the *SNORD116* cluster genes and several other genes such as *PWAR1*, 5 and 6, *PWARSN* and *IPW*. In addition to the PWS/AS block, we detected six other PofO methylation bias blocks ranging from 35 to 65 kb in size, were located within *ZNF331*, *KCNQ1OT1*, *GNAS/GNAS-AS1*, *L3MBTL1*, *ZNF597/NAA60*, and *GPR1-AS/ZDBF2* imprinted clusters (*Figure 8—figure supplements 1–6*).

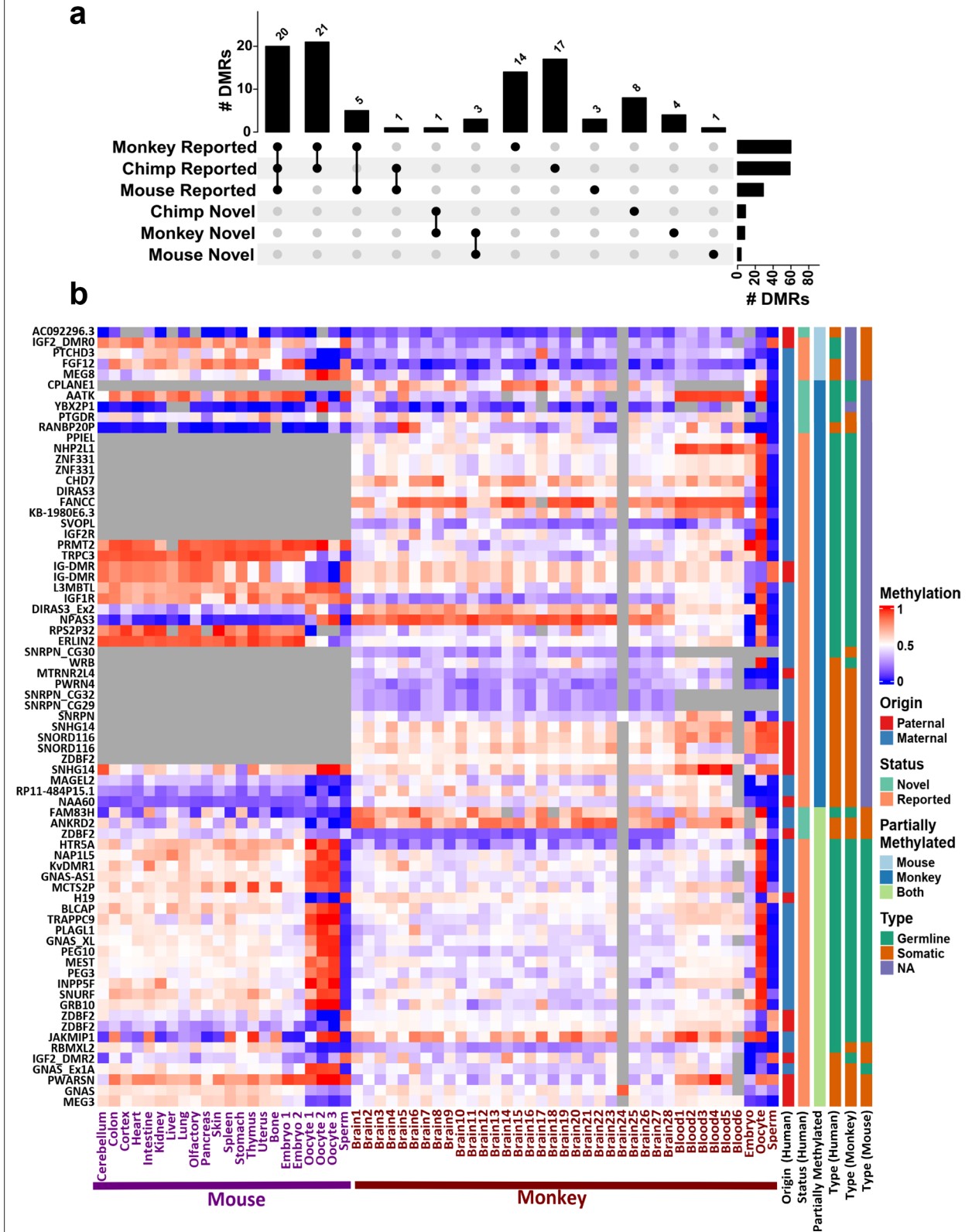

**Figure 5.** Conservation of detected differentially methylated regions (DMRs). (**a**) Upset plot representing the number of previously reported and novel DMRs with evidence of conservation (partial methylation) in each of the mammals. (**b**) Heatmap representing human DMRs (DMR names on the left) and average methylation of their orthologous intervals in mouse and macaque in different tissues and also in sperm, oocyte, and embryonic samples. Gray regions represent *NA* values that either did not have an ortholog or enough CpG in the whole-genome bisulfite sequencing (WGBS) data.

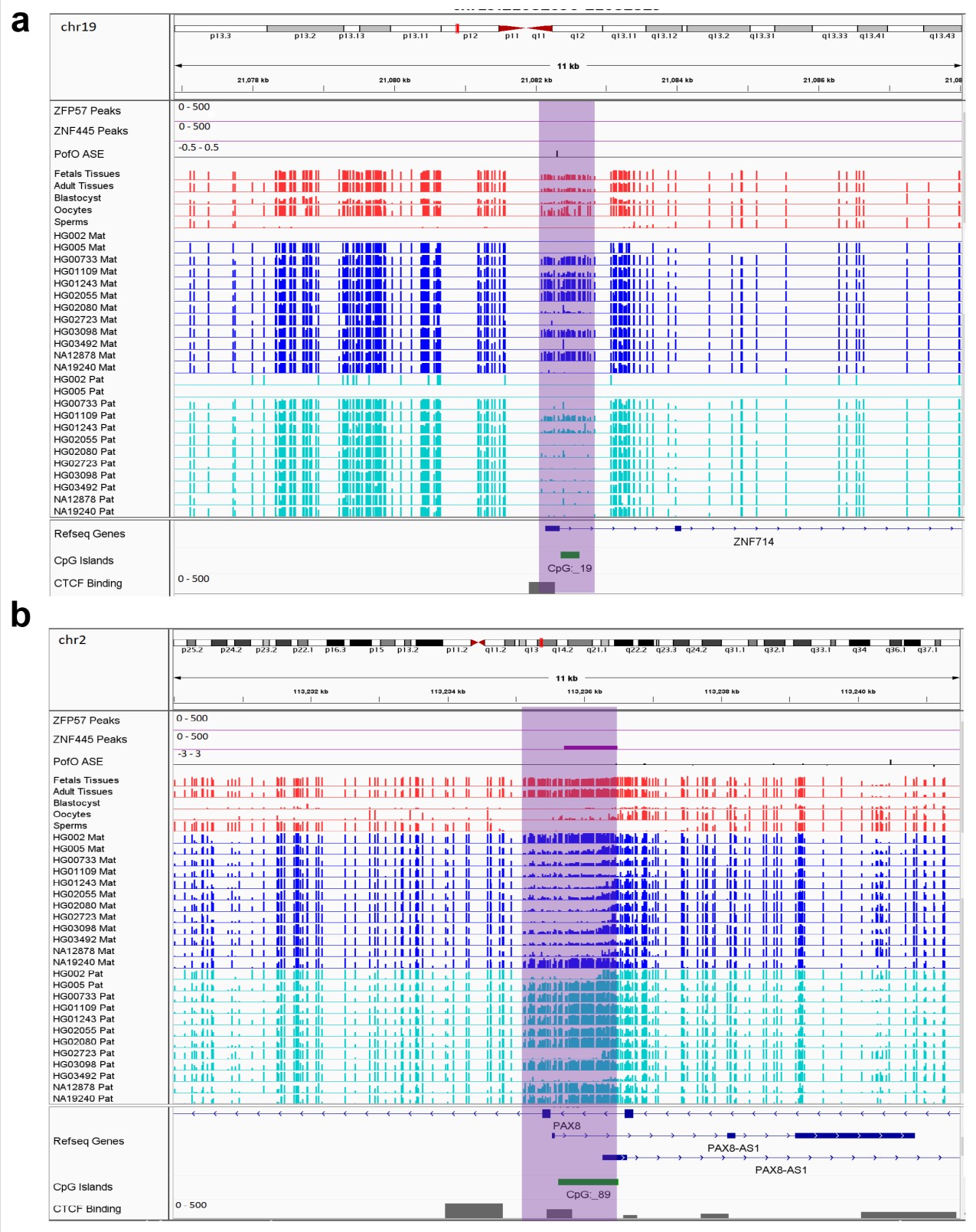

**Figure 6.** IGV screenshots of two novel somatic differentially methylated regions (DMRs). (**a**) Novel maternally methylated somatic DMR overlapping with the promoter of paternally expressed *ZNF714* gene. (**b**) Novel paternally methylated somatic DMR overlapping with the promoter of maternally expressed *PAX8-AS1* gene. Highlighted box regions represent the DMRs. Parent-of-origin (PofO) allele-specific expression (ASE) track is created using

*Figure 6 continued on next page*

*Figure 6 continued*

publicly available ASE data from *Zink et al., 2018* (see Materials and methods). Positive vertical bars (upward) represent paternal expression and negative bars (downward) represent maternal expression. The range for all methylation tracks is 0–1.

The online version of this article includes the following figure supplement(s) for figure 6:

**Figure supplement 1.** Novel somatic paternally methylated differentially methylated region (DMR) in paternally expressed *ZDBF2* gene.

**Figure supplement 2.** Novel somatic maternally methylated differentially methylated region (DMR) in maternally expressed *RB1* gene and isoform dependent (or in some studies paternally) expressed *LPAR6* gene.

**Figure supplement 3.** Novel somatic maternally methylated differentially methylated region (DMR) in paternally expressed *BMP8A* gene.

**Figure supplement 4.** Novel somatic paternally methylated differentially methylated region (DMR) 3 kb away from paternally expressed *PWAR1* gene.

**Figure supplement 5.** Novel somatic paternally methylated differentially methylated region (DMR) 90 kb away from maternally expressed *LINC00665* gene.

**Figure supplement 6.** Novel somatic paternally methylated differentially methylated region (DMR) 296 kb away from randomly/maternally expressed *DGCR6* gene.

**Figure supplement 7.** Novel somatic paternally methylated differentially methylated region (DMR) 1.03 Mb away from maternally expressed *IGF2R*.

As mentioned in the 'Confirmation of novel imprinted DMRs' section, only 42 out of 99 detected novel DMRs in the nanopore data could be confirmed in the WGBS data as partially methylated. Forty of the novel nanopore-detected DMRs that did not show partial methylation in the WGBS data mapped to the seven PofO-biased blocks. At imprinted intervals one allele is methylated and the other one is not. Therefore, at these intervals aggregated methylation from both alleles demonstrate partial methylation (~50% methylation) in WGBS data. However, in the subtle PofO bias blocks both alleles are methylated with a subtle hypomethylation on one of the alleles. Therefore, in contrast to imprinted intervals, aggregated methylation at these blocks usually do not show partial methylation in WGBS data. The weaker or subtle differential methylation can therefore explain why several novel DMRs detected in the nanopore data did not show partial methylation in the WGBS data and demonstrates the utility of nanopore sequencing in detecting subtle ASM differences.

## Enriched allelic H3K36me3 and H3K27me3 histone marks at contiguous blocks

RNA polymerase II recruits SETD2 during elongation which results in the deposition of the H3K36me3 mark in the gene body. In turn, H3K36me3 recruits de novo DNA methyltransferases through their PWWP domain which results in DNA methylation in the gene body (*Wagner and Carpenter, 2012*).

Within the seven PofO methylation-biased blocks, parentally expressed or active allele demonstrated hypermethylation suggesting that subtle methylation is linked to parental ASE. Except *ZNF597/NAA60*, all the blocks demonstrated hypermethylation and ASE on the paternal allele. *ZNF597/NAA60* demonstrated hypermethylation and ASE on the maternal allele. Therefore, to assess allelic H3K36me3, we used ChIP-seq data from six LCLs (*Kasowski et al., 2013*). H3K36me3 and H3K27me3 histone marks are mutually exclusive (*Yuan et al., 2011*). Moreover, DNA methylation and H3K27me3 shown to be mutually exclusive at CGIs (*Brinkman et al., 2012*). Thus, we also examined allelic H3K27me3 in the same cell line samples (*Kasowski et al., 2013*).

To analyze allelic histone modifications and detect blocks of allelic histone marks at large blocks of PofO bias, we binned the genome into 10 kb intervals and performed a binomial test with Fisher's combined p-value test to determine the significance of allelic read counts at 10 kb intervals with >3 informative heterozygous SNVs (having at least five mapped reads) within each block in each sample. A 10 kb bin considered as significant for allelic histone mark if it had an adjusted p-value <0.001 and if at least 70% of the SNVs within the 10 kb bin having ≥80% of the reads mapped to one allele. In total, 174 bins for H3K36me3 and 132 bins for H3K27me3 could be examined. Of these, 147 bins for H3K36me3 and 51 bins for H3K27me3 were significant. Thirty-eight bins were significant for both histone marks in the same sample. All the seven blocks demonstrated multiple significant bins for H3K36me3 at almost all the samples. *L3MBTL1*, *GPR1-AS/ZDBF2*, *GNAS/GNAS-AS1*, and *ZNF597/NAA60* demonstrated multiple significant H3K27me3 bins in majority of the samples and *KCNQ1OT1*, *PWS/AS*, and *ZNF331* had significant H3K27me3 bins at 3, 2, and 1 of the samples, respectively. H3K36me3 and H3K27me3 demonstrated mutual exclusive pattern and H3K36me3 appeared on the

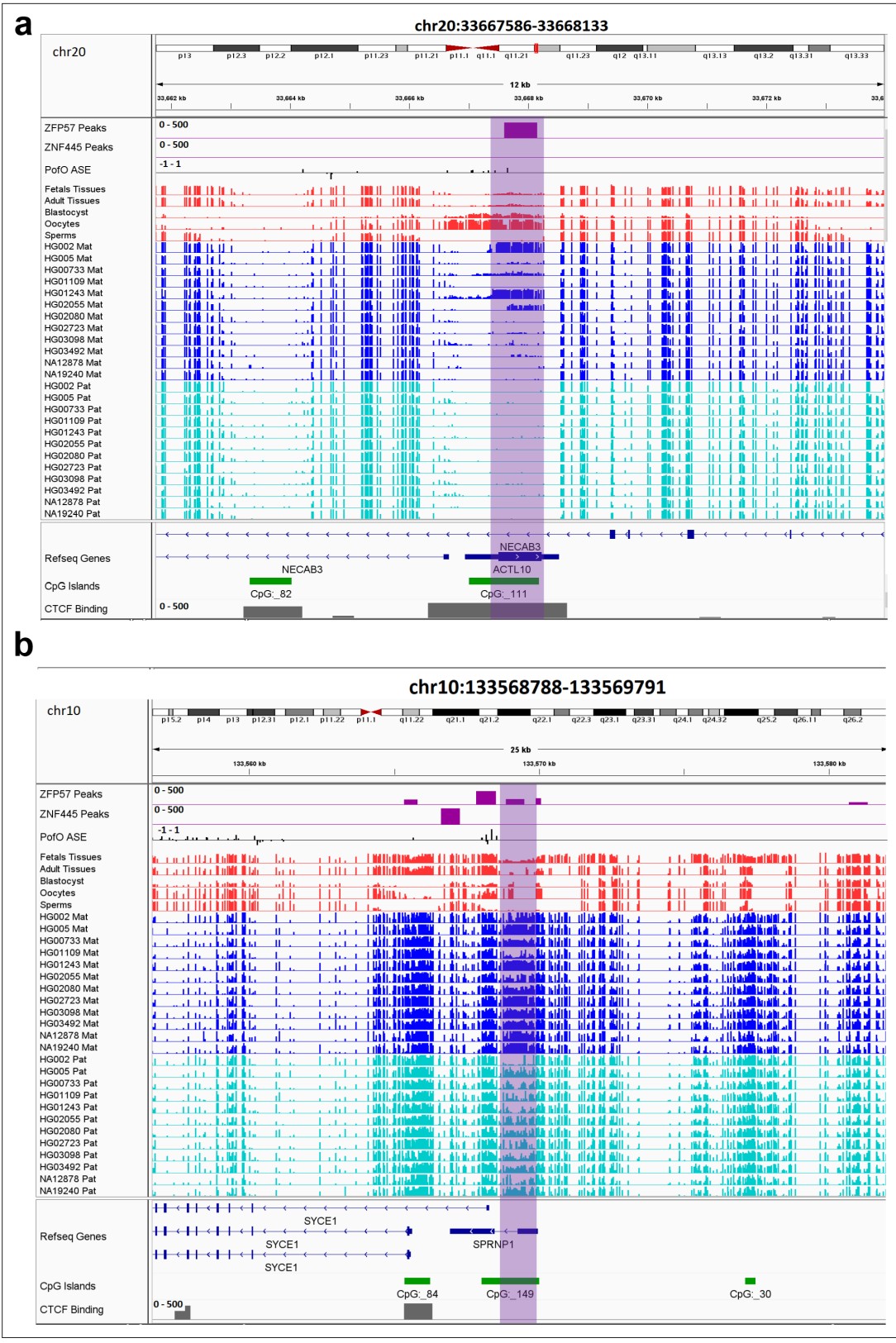

**Figure 7.** IGV screenshots of two novel maternal germline differentially methylated regions (DMRs). (**a**) Novel maternally methylated germline DMR overlapping with the promoter of the paternally expressed *ACTL10* gene. (**b**) Novel maternally methylated germline DMR overlapping with the promoter of the *SYCE1* gene, which demonstrates paternal expression bias from parent-of-origin (PofO) allele-specific expression (ASE) track. Highlighted box regions represent the DMRs. PofO ASE track is created using publicly available ASE data from *Zink et al., 2018* (see Materials and

*Figure 7 continued on next page*

*Figure 7 continued*

methods). Positive vertical bars (upward) represent paternal expression and negative bars (downward) represent maternal expression. The range for all methylation tracks is 0–1.

The online version of this article includes the following figure supplement(s) for figure 7:

**Figure supplement 1.** Novel germline maternally methylated differentially methylated region (DMR) in maternally expressed *DDA1* gene.

**Figure supplement 2.** Novel germline maternally methylated differentially methylated region (DMR) in paternally expressed *AC024940.1* (*OVOS2*) gene.

**Figure supplement 3.** Novel germline maternally methylated differentially methylated region (DMR) 149 kb away from isoform dependent expressed *NAPRT* gene.

**Figure supplement 4.** Novel germline maternally methylated differentially methylated region (DMR) 745 kb away from maternally expressed *NTM* gene.

hypermethylated allele while H3K27me3 on the hypomethylated allele (*Figure 8*; *Figure 8—figure supplements 1–6*; *Figure 9*; *Supplementary file 8*).

To determine if allelic histone marks are unique to the PofO methylation-biased blocks, we examined allelic histone marks on several other imprinted clusters with strong ASE which did not display PofO bias methylation. For this, we examined *PPIEL*, *MEG3*, *MEST*, *DIRAS3*, *IGF2*, *MTRNR2L4*, and *ADNP2/PARD6G-AS1* clusters. Eighty-three bins for H3K36me3 and 138 bins for H3K27me3 could be examined at the seven test blocks. Of these, only five bins for H3K36me3 and seven bins for H3K27me3 were significant and none of the bins were significant for both histone marks (*Figure 9—figure supplements 1–8*; *Supplementary file 9*). These results suggest that the blocks of PofO methylation bias in the gene body of active alleles are mediated by transcription and histone marks at their gene bodies.

## Discussion

Here, we describe the first genome-wide map of human ASM intended to detect novel imprinted intervals using nanopore sequencing. Leveraging long reads and parental SNVs allowed us to phase methylation for ~26.5 million autosomal CpGs representing 95% of the CpGs in the human autosomal genome (GRCh38) across 12 LCLs. This effort achieves a much higher resolution than previous studies aimed at capturing allelic methylation using bisulfite sequencing or methylation arrays (*Court et al., 2014*; *Hernandez Mora et al., 2018*; *Joshi et al., 2016*; *Zink et al., 2018*). Fourteen of our novel DMRs did not have any phased CpG from previous WGBS or array studies (*Supplementary file 4*), illustrating the utility of longer reads for imprinted methylation calling. DMRs that are detected in only a single study displayed higher variations across individuals compared to those detected by at least two studies. Therefore, lack of phasing at some novel DMRs in previous studies and higher variation in imprinted methylation at novel DMRs could explain the reason they were not detected previously. We also demonstrated that germline DMRs with a greater number of *ZFP57* motif tend to be more consistently imprinted across individuals suggesting motifs redundancy increases *ZFP57* recruitment and tolerance to any DNA sequence variation. However, due to the availability of DNA sequence in a limited number of samples, we were not able to examine sequence variation at the DMRs and the *ZFP57*-binding motifs for any possible association with polymorphic imprinted methylation which will require further study.

Even though we detected methylation for all the CpGs in the human genome (GRCh38), we were not able to phase 5% of the human methylome (*Kent et al., 2002*). We used SNVs detected from short-reads data in the 1KGP and GIAB databases for phasing (*Auton et al., 2015*; *Zook et al., 2019*). Seventy-five percent of the unphased CpGs mapped to the ENCODE blacklist, regions with low mappability, indicative of lack of SNVs to phase reads (*Amemiya et al., 2019*). Improvement in base calling and variant calling from nanopore reads could enable the phasing of a complete genome-wide methylome using nanopore-detected SNVs.

We detected 16 novel gDMRs and 26 novel sDMRs. These novel DMRs were supported by several lines of evidence in our analyses. (1) They displayed significant PofO methylation bias in nanopore-sequenced cell line samples. (2) They were partially methylated in WGBS data. (3) gDMRs demonstrated establishment of methylation in sperm or oocyte and escape from the second de-methylation

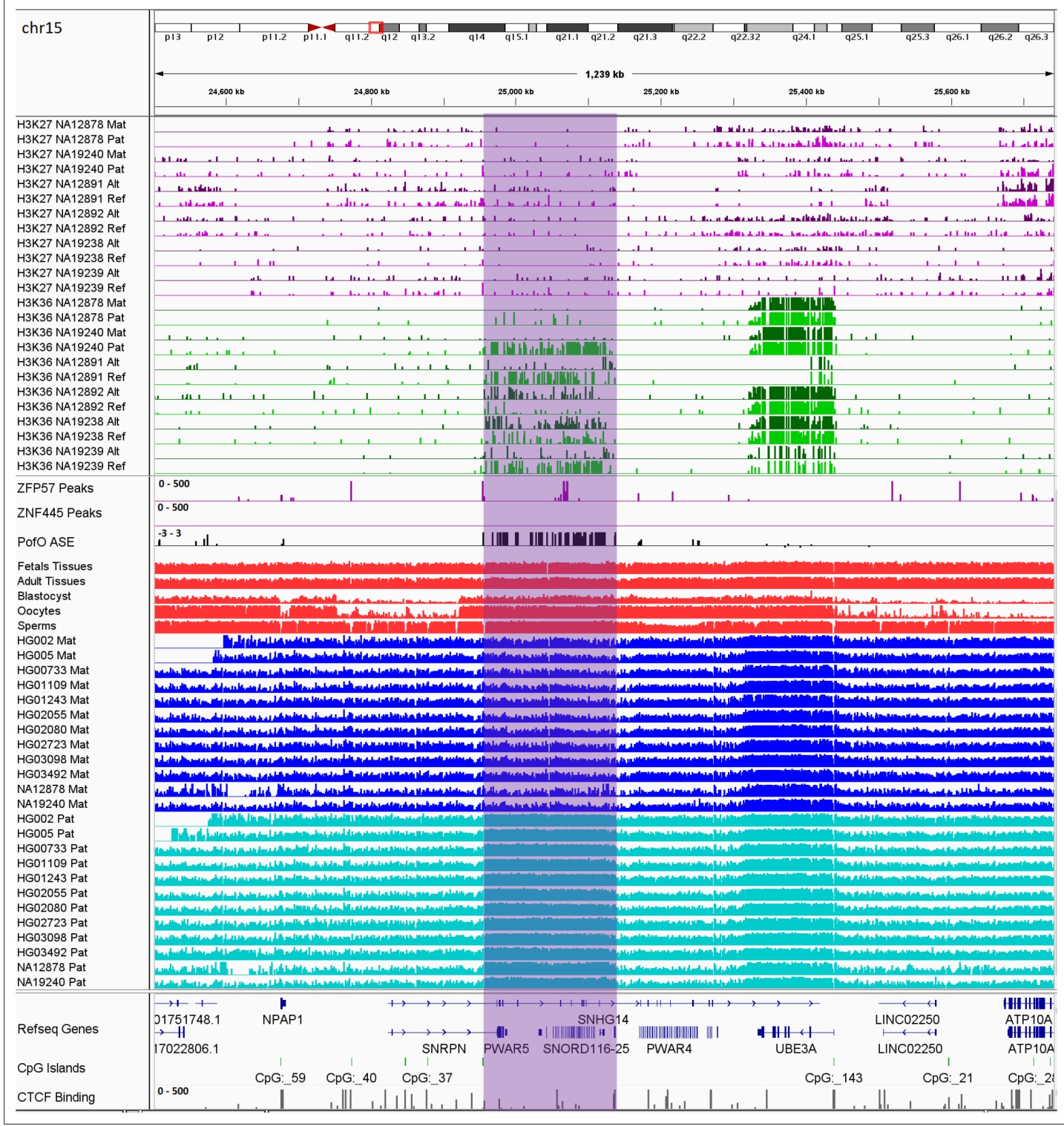

**Figure 8.** IGV screenshot of 200 kb paternally methylated biased methylation block in the PWS/AS imprinted cluster. The range for all methylation tracks is 0–1. The histone mark tracks represent allelic read counts for H3K36me3 and H3K27me3 modifications. The range for all histone mark tracks is 0–20. In H3K27 and H3K36 tracks, for NA12878 and NA19240 the parent-of-origin could be determined and specified by maternal (Mat) and paternal (Pat) alleles. While the other samples are specified by reference (Ref) and alternative (Alt) alleles.

The online version of this article includes the following figure supplement(s) for figure 8:

**Figure supplement 1.** IGV screenshot of ~65 kb paternally methylated biased methylation block in *GPR1-AS/ZDBF2* imprinted cluster.

step. (4) Eighty-four percent of those for which H3K4me3 ChIP-seq data could be phased and examined showed significant allelic H3K4me3. (5) Forty percent showed evidence of conservation. (6) Eighty-three percent mapped to at least one regulatory region including CGI, CTCF-binding site, and enhancer (*Supplementary file 4*). These novel DMRs represent a substantial and well-confirmed expansion of known regions of imprinting, which may aid future research and diagnosis in the fields of genetic medicine and oncology.

We detected seven blocks of allelic methylation bias (*Figure 8*; *Figure 8—figure supplements 1–6*). All of the blocks represented several common features. (1) They were detected in imprinted genes that appeared in a cluster. (2) There was at least one well-characterized and conserved gDMR in each block (except *ZNF597/NAA60* block with a conserved sDMR). (3) The well-characterized DMRs in these blocks displayed significant allelic H3K4me3 (except the DMR in the *L3MBTL1* block, which could not be examined due to the lack of an SNV). (4) The well-characterized DMRs in these blocks overlapped with the promoters of genes with subtle PofO methylation bias at the gene body and DMR itself displayed opposite PofO methylation (except for *GPR1-AS/ZDBF2* block where DMR did not map to the promoter and had the same PofO with the gene body). (5) All the blocks were accompanied by a strong allelic expression and H3K36me3 histone mark on the subtle hypermethylated allele and H3K27me3 on the hypomethylated allele. This represents a novel facet of imprinting biology and suggests a link between allelic expression and histone modifications with biased PofO methylation at these blocks. However, the mechanism regulating such blocks and the rule of these PofO-biased methylation remain to be determined. One possible explanation could be that the subtle parental methylation bias is used by cells to express important genes (genes that can regulate other genes in the cluster or have regulatory roles) in an imprinted cluster with higher fidelity through its gene body methylation on the active allele. For example, at the *KCNQ1OT1* and *GNAS/GNAS-AS1* clusters, the methylation blocks overlap with *KCNQ1OT1* and *GNAS-AS1* gene bodies, both of which encode antisense RNA transcripts that regulate other genes in the imprinted cluster (*Chiesa et al., 2012*; *Turan and Bastepe, 2013*).

Orthologous regions of ~40% of the detected DMRs demonstrated partial methylation in one or more of the three mammals including chimpanzee, rhesus macaque, and mouse, suggesting their conservation. There were a considerably higher number of orthologous sites and partially methylated orthologous DMRs in chimpanzee and rhesus macaque, in agreement with more similarities and less distance to these primates compared to the mouse in human evolution. Previously, *Court et al., 2014*, detected 14 novel DMRs, and did not detect any imprinted orthologs of their novel DMRs in mice (*Court et al., 2014*). All 14 DMRs also overlapped with our detected DMRs and six of them had orthologous regions in mm10 using the UCSC liftover file (*Kent et al., 2002*). Two of the orthologs displayed partial methylation in mouse; the first is *MEG8* human DMR with its orthologous *Rian* gene in the mouse, which was not examined by *Court et al., 2014*, and the other is found in the *Htr5a* gene, which was previously reported as not conserved in mouse (*Court et al., 2014*). After reviewing their analysis, *Court et al., 2014*, seem to have examined different orthologous region (*Appendix 2—figure 1*). For *Htr5a*, they examined the CGI (CpG:_102) ~50 kb away from the gene, while we examined the region spanning the first or second exon (two transcripts) of *Htr5a* which was partially methylated while CpG:_102 was also unmethylated in our study.

Using reported imprinted genes, 36% of the novel DMRs mapped close to known imprinted genes (*Babak et al., 2015*; *Baran et al., 2015*; *Geneimprint, 2021*; *Jadhav et al., 2019*; *Morison et al., 2001*; *Zink et al., 2018*). Five of our novel DMRs could be potential ICRs for reported imprinted genes. Specifically, imprinted DMRs overlapping the promoters of *ZNF714*, *PAX8-AS1*, *ACTL10,* and *SYCE1* genes (*Figures 6 and 7*). *ZNF714* is a member of the zinc finger family of proteins which have several imprinted genes with developmental roles (*Babak et al., 2015*; *Baran et al., 2015*; *Camargo*

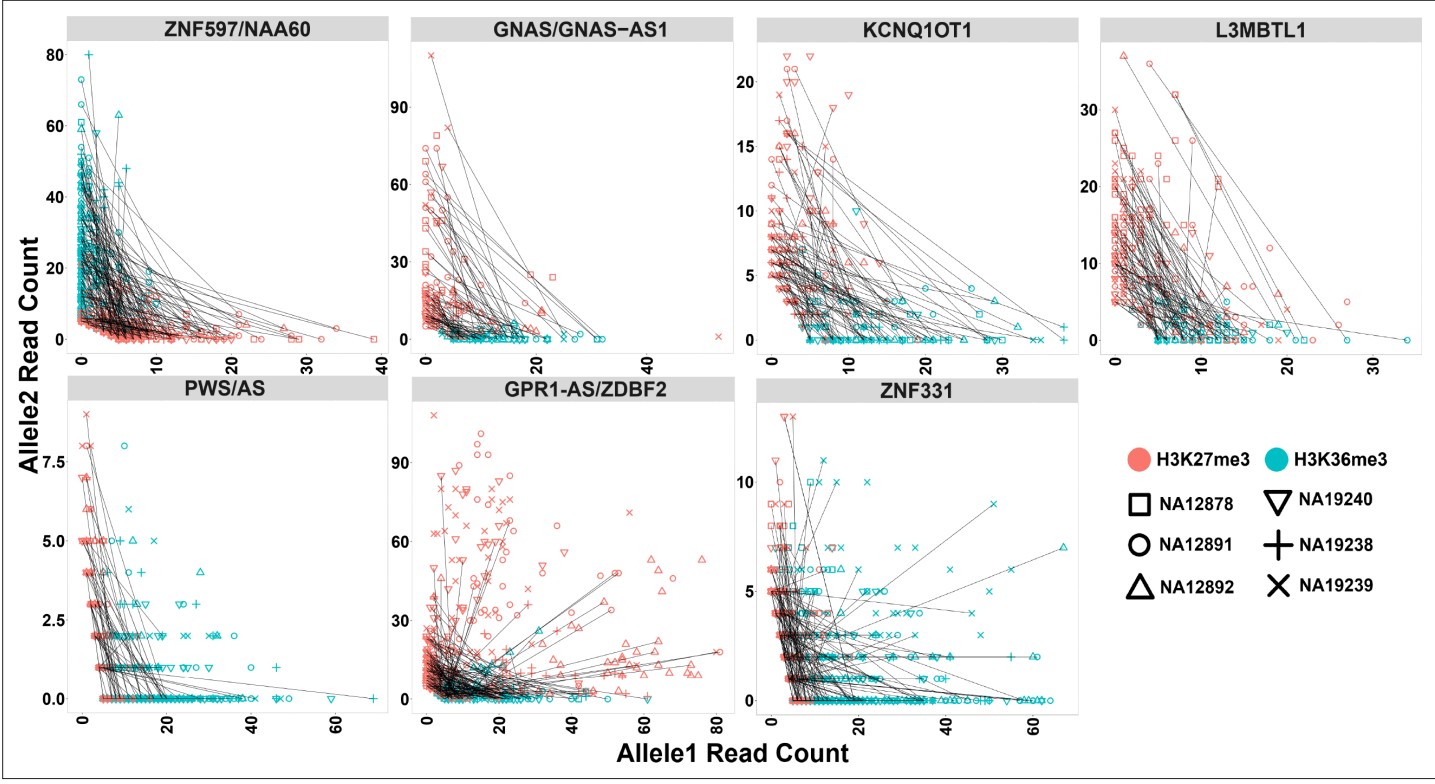

**Figure 9.** Mutual exclusive allelic H3K36me3 and H3K27me3 histone marks at seven parent-of-origin (PofO) methylation-biased blocks. All blocks demonstrate allelic H3K36me3 on hypermethylated allele and H3K27me3 on hypomethylated allele. For NA12878 and NA19240, allele1 is the paternal and allele2 is maternal. For sake of visualization in other four cell lines without parental information, allele1 for H3K36me3 mark demonstrates the allele with more mapped reads at all blocks except *ZNF597/NAA60*. Therefore, for H3K36me3 we swapped the reference allele read count with the alternative allele read count if the reference allele count was less than the alternative allele count. At *ZNF597/NAA60*, we swapped the reference allele read count with the alternative allele read count if the reference had higher read count. We also swapped the reference and the alternative allele counts for the same SNVs for H3K27me3. Each point represents a heterozygous SNV. Lines are connecting SNVs that have mapped reads for both histone modifications.

The online version of this article includes the following source data and figure supplement(s) for figure 9:

**Source data 1.** H3K36me3 and H3K27me3 allelic read counts for the heterozygous single-nucleotide variants (SNVs) mapped to the detected PofO-biased methylation blocks.

**Figure supplement 1.** Allelic H3K36me3 and H3K27me3 histone marks read count at seven test blocks.

**Figure supplement 1—source data 1.** H3K36me3 and H3K27me3 allelic read counts for the heterozygous single-nucleotide variants (SNVs) mapped to the test blocks.

**Figure supplement 2.** IGV screenshot of the *PPIEL* imprinted cluster.

**Figure supplement 3.** IGV screenshot of the *MEG3* imprinted cluster.

**Figure supplement 4.** IGV screenshot of the *MEST* imprinted cluster.

**Figure supplement 5.** IGV screenshot of the *DIRAS3* imprinted cluster.

**Figure supplement 6.** IGV screenshot of the *IGF2* imprinted cluster.

**Figure supplement 7.** IGV screenshot of the *MTRNR2L4* imprinted cluster.

**Figure supplement 8.** IGV screenshot of the *ADNP2* imprinted cluster.

*et al., 2012*; *Jadhav et al., 2019*; *Zink et al., 2018*). *ZNF714* has been reported to be associated with non-syndromic cleft lip (*Camargo et al., 2012*). Aberrant methylation of multiple CpGs overlapping with the novel DMR at *PAX8-AS1* has been implicated in thyroid disorders (*Candler et al., 2021*). *SYCE1* and *ACTL10* are also implicated in human diseases (*Bak et al., 2016*; *Maor-Sagie et al., 2015*). Thus, these imprinted DMRs could be of potential clinical value.

In addition to the aforementioned novel DMRs, two of the reported DMRs in *PTCHD3* and *FANCC* are also interesting. Paternal expression of *PTCHD3* and maternal expression for *FANCC* were

previously detected by *Zink et al., 2018*, though they could not detect any associated DMR due to the lack of phased CpG (*Zink et al., 2018*). *Hernandez Mora et al., 2018*, detected three maternally methylated probes at the promoter of *PTCHD3* and one maternally methylated probe in intron 1 of *FANCC*, but were unable to examine the parental expression (*Hernandez Mora et al., 2018*). We also detected two maternally methylated gDMRs overlapping with the promoter of *PCTHD3* and intron 1 of *FANCC* (*Appendix 3—figure 1*; *Appendix 3—figure 2*). Therefore, these gDMRs could be the ICRs for these genes. The maternal gDMR at the *PTCHD3* promoter can directly suppress the maternal allele and results in paternal expression. *FANCC* gDMR overlaps with a CGI and CTCF-binding site. CTCF is a methylation-sensitive DNA-binding protein and CpG methylation can inhibit CTCF binding (*Hashimoto et al., 2017*; *Renda et al., 2007*). Moreover, CTCF binding to the first intron of major immediate early (MIE) gene of the human cytomegalovirus (HCMV) in HCMV-infected cells resulted in repression of this gene (*Puerta et al., 2014*). Therefore, the maternally methylated DMR in intron 1 of maternally expressed *FANCC* suggests a mechanism through which the paternal allele is suppressed by CTCF binding at DMR while DNA methylation inhibits CTCF binding at the maternal allele.

Overall, our study demonstrates a near-complete genome-wide map of human ASM by leveraging long-read nanopore technology. The use of nanopore technology allowed us to expand the set of known imprinted DMRs using 12 LCLs with parental SNPs. Moreover, we detected seven large PofO bias methylation blocks with enriched allelic expression and histone modifications. We showed that nanopore sequencing has the ability to achieve a higher resolution of phased CpGs using a small sample size and allows for the calling of imprinted methylation in a single sample, potentially reducing the cost by reducing the sample size.

# Materials and methods

## Nanopore sequencing data and detection of ASM

We used publicly available nanopore sequencing data for 12 LCLs with trio data available. Raw and base-called nanopore data for HG002, HG005, HG00733, HG01109, HG01243, HG02055, HG02080, HG02723, HG03098, and HG03492 were obtained from the Human Pangenomics and GIAB (*Shafin et al., 2020*; *Zook et al., 2016*). NA19240 data (ERR3046934 and ERR3046935 raw nanopore and their base-called reads ERR3219853 and ERR3219854) were obtained from *De Coster et al., 2019*. Raw and base-called nanopore data for NA12878 were obtained from rel6 nanopore WGS consortium (*Jain et al., 2018*). Reads were mapped to the GRCh38 human reference genome using Minimap2 with the setting *minimap2 –ax map-ont* (*Kent et al., 2002*; *Li, 2018*). For all the cell lines and their parents, except HG002 and HG005, high-quality SNVs were called using Strelka2 with default parameters from alignment files in the 1KGP GRCh38 (*Auton et al., 2015*; *Kim et al., 2018*). High-quality SNVs for HG002 and HG005 and their parents were obtained from GIAB v.3.3.2 high confidence variant calls (*Zook et al., 2019*). CpG methylations were called from nanopore data using nanopolish with default parameters (*Simpson et al., 2017*). Methylation calls for each sample were preprocessed using the NanoMethPhase *methyl_call_processor* default setting (*Akbari et al., 2021*). Subsequently, haplotyping and PofO methylation detection were performed using NanoMethPhase and trio (mother, father, and child) variant call data with the setting *nanomethphase phase –mbq 0*. Finally, DMRs between haplotypes were called using the default setting of NanoMethPhase *dma* module that uses Dispersion Shrinkage for Sequencing data R package for DMA (*Park and Wu, 2016*). To avoid the confounding effects of X-chromosome inactivation, and because previous studies demonstrated no evidence of imprinting at sex chromosomes, we only examined autosomal chromosomes (*Court et al., 2014*; *Joshi et al., 2016*; *Zink et al., 2018*).

## WGBS data and detection of novel DMRs

To confirm allelic methylation in other tissues and also detect potential novel imprinted DMRs, we used 60 public WGBS data records for 29 tissue type samples from the Epigenomics Roadmap and ENCODE projects (*Supplementary file 10*) and 119 blood WGBS datasets for 87 individuals from the Blueprint project (*Bernstein et al., 2010*; *ENCODE Project Consortium, 2012*; *Stunnenberg et al., 2016*; *Supplementary file 10*). CpGs with at least five mapped reads were used for further analysis. At imprinted DMRs, only one allele is methylated and we expect to observe partial methylation (~50%) at such regions. Therefore, we investigated the partial methylation of nanopore-detected DMRs in

WGBS data (code is available on https://github.com/vahidAK/NanoMethPhase/tree/master/scripts (*Akbari, 2022*): PartialMethylation_AtDMR.sh). As controls, we examined 100 randomly selected CGIs: 1, 2, and 3 kb intervals with more than 15 CpGs each resampled 100 times.

### Detection of gDMRs and sDMRs

To discriminate gDMRs from sDMRs, we used publicly available WGBS data for three sperms, two oocytes, and one blastocyst first published by *Okae et al., 2014*, and three fetal tissue libraries (GSM1172595 thymus, GSM1172596 muscle, and GSM941747 brain) from the Roadmap project (*Bernstein et al., 2010*; *Okae et al., 2014*).

### Allelic H3K4me3, H3K36me3, and H3K27me3 analysis

H3K4me3, H3K36me3, and H3K27me3 ChIP-seq fastq files were obtained for NA12878, NA12891, NA12892, NA19238, NA19239, and NA19240 (SRP030041) (*Kasowski et al., 2013*). ChIP-seq data were aligned to the GRCh38 reference genome using the bwa-mem default setting (*Kent et al., 2002*; *Li and Durbin, 2009*). High-quality SNVs were called for these samples from 1KGP GRCh38 alignment files using strelka2 (*Auton et al., 2015*; *Kim et al., 2018*). We then counted the number of reads with a minimum mapping quality of 20 and base quality of 10 at each heterozygous SNV and kept those with at least five mapped reads. The reference allelic counts and total counts at each heterozygous SNV were used to detect significant allelic bias using a two-sided binomial test under the default probability of $p=0.5$ in python SciPy package (codes are available on GitHub https://github.com/vahidAK/NanoMethPhase/tree/master/scripts: CountReadsAtSNV.py & Binomial_test.py) (*Virtanen et al., 2020*).

### ASE track

ASE data from *Zink et al., 2018* (PofO_ASE.tsv; https://doi.org/10.6084/m9.figshare.6816917) were used to create ASE track for IGV. In PofO_ASE.tsv file from Zink et al., they have calculated lor_paternal_maternal across individuals which is (lor_ref_alt_pref - lor_ref_alt_palt)/2. lor_ref_alt_pref is log(#reads with ref allele/#reads with alt allele) when paternal homologue has ref allele and lor_ref_alt_palt is log(#reads with ref allele/#reads with alt allele) when paternal homologue has alt allele. For visualization in IGV, we converted the PofO_ASE.tsv file from Zink et al., to a bigwig format file using the UCSC tool bedGraphToBigWig version 4 and we kept lor_paternal_maternal as ASE value (*Kent et al., 2010*).

### Mammalian conservation of DMRs

We used 16 WGBS datasets for mouse (GSM1051150-60 and GSM1051162-66), 34 WGBS datasets for rhesus macaque (GSE34128 and GSE151768), and 22 WGBS datasets for chimpanzee (GSE151768) to examine partial methylation in orthologous intervals (*Hon et al., 2013*; *Jeong et al., 2021*; *Tung et al., 2012*). Mouse, macaque, and chimpanzee coordinates lifted over to mm10, RheMac8, and PanTro5 coordinates using CrossMap and the appropriate liftover file from the UCSC genome browser. The list of detected human DMRs were also converted to the orthologous regions for each mammal using CrossMap and the appropriate liftover file (*Kent et al., 2002*; *Zhao et al., 2014*). Since many coordinates in the human splitted to several orthologs in other mammals, we merged orthologs that were ≤200 bp apart.

To examine the somatic and germline ortholog DMRs, we used embryo (GSM3752614, GSM4558210), sperm (GSE79226), and oocyte (GSM3681773, GSM3681774, GSM3681775) WGBS libraries from mouse; and embryo (GSM1466814), sperm (GSM1466810), and oocyte (GSM1466811) WGBS libraries from rhesus macaque (*Dahlet et al., 2020*; *Gao et al., 2017*; *Jung et al., 2017*; *Saenz-de-Juano et al., 2019*).

## Acknowledgements

SJMJ and MAM acknowledge funding from the Canada Research Chairs program and the Canadian Foundation for Innovation. VA acknowledges funding from the University of British Columbia with a Four-Year Doctoral Fellowship.

## Additional information

### Funding

| Funder | Grant reference number | Author |
|---|---|---|
| The University of British Columbia, 4-Year Doctoral Fellowship | | Vahid Akbari |
| Canada Research Chairs | | Marco A Marra<br>Steven JM Jones |

The funders had no role in study design, data collection and interpretation, or the decision to submit the work for publication.

### Author contributions

Vahid Akbari, Conceptualization, Data curation, Formal analysis, Funding acquisition, Investigation, Methodology, Software, Validation, Visualization, Writing - original draft, Writing – review and editing; Jean-Michel Garant, Conceptualization, Data curation, Software, Writing – review and editing; Kieran O'Neill, Conceptualization, Data curation, Formal analysis, Investigation, Software, Writing – review and editing; Pawan Pandoh, Conceptualization, Formal analysis, Investigation, Writing – review and editing; Richard Moore, Conceptualization, Formal analysis, Writing – review and editing; Marco A Marra, Martin Hirst, Conceptualization, Formal analysis, Resources, Writing – review and editing; Steven JM Jones, Conceptualization, Funding acquisition, Project administration, Resources, Supervision, Writing – review and editing

### Author ORCIDs

Vahid Akbari http://orcid.org/0000-0001-8005-7776
Kieran O'Neill http://orcid.org/0000-0001-7609-5905
Marco A Marra http://orcid.org/0000-0001-7146-7175
Steven JM Jones http://orcid.org/0000-0003-3394-2208

### Decision letter and Author response

Decision letter https://doi.org/10.7554/eLife.77898.sa1
Author response https://doi.org/10.7554/eLife.77898.sa2

## Additional files

### Supplementary files

• Appendix 1—figure 1—source data 1. Number of ZFP57-binding motif (TGCCGC) at differentially methylated regions (DMRs).

• Supplementary file 1. List of the reported imprinted differentially methylated regions (DMRs).

• Supplementary file 2. The list of 200 detected imprinted differentially methylation regions from nanopore data. This list demonstrates the results of differential methylation analysis (DMA) using nanomethphase dma module that uses DSS R package for DMA.

• Supplementary file 3. Mapping detected differentially methylated regions (DMRs) using nanopore sequencing across 12 lymphoblastoid cell lines (LCLs) to reported DMRs.

• Supplementary file 4. Detected differentially methylated regions (DMRs) using nanopore sequencing across 12 lymphoblastoid cell lines (LCLs) that mapped to reported DMRs or confirmed in whole-genome bisulfite sequencing (WGBS) data.

• Supplementary file 5. The results for examining inter-individual variations of the detected differentially methylated regions (DMRs) in whole-genome bisulfite sequencing (WGBS) datasets from blood.

• Supplementary file 6. The allelic read counts of H3K4me3 chromatin immunoprecipitation sequencing (ChIP-seq) for each heterozygous single-nucleotide variant (SNV) at the detected differentially methylated regions (DMRs) in six lymphoblastoid cell line (LCL) samples.

• Supplementary file 7. List of the known imprinted genes.

• Supplementary file 8. The allelic read counts of H3K36me3 and H3K27me3 chromatin

immunoprecipitation sequencing (ChIP-seq) for the seven parent-of-origin (PofO)-biased methylation blocks in six lymphoblastoid cell line (LCL) samples.

• Supplementary file 9. The allelic read counts of H3K36me3 and H3K27me3 chromatin immunoprecipitation sequencing (ChIP-seq) for the seven test blocks in six lymphoblastoid cell line (LCL) samples.

• Supplementary file 10. List of the whole-genome bisulfite sequencing (WGBS) tissues and WGBS blood samples used in our study.

• Transparent reporting form

### Data availability

The current manuscript is a computational study, so no new datasets have been generated for this manuscript. The source of each dataset is provided under the '"Materials and methods'" section under the appropriate subsection. Genomic tracks generated in this study including DNA methylation and histone modification tracks are deposited in the Mendeley data repository (https://doi.org/10.17632/f4k2gytbh5.1). Codes are uploaded to GitHub https://github.com/vahidAK/NanoMethPhase/tree/master/scripts (copy archived at swh:1:rev:1657f7aed60604aa7c7f3e77d992d76bee6bf6d3): PartialMethylation_AtDMR.sh, CountReadsAtSNV.py and Binomial_test.py.

The following dataset was generated:

| Author(s) | Year | Dataset title | Dataset URL | Database and Identifier |
|---|---|---|---|---|
| Akbari V | 2022 | Genome-Wide Detection of Imprinted Differentially Methylated Regions Using Nanopore Sequencing_ Akbari-etal | https://doi.org/10.17632/f4k2gytbh5.1 | Mendeley Data, 10.17632/f4k2gytbh5.1 |

The following previously published datasets were used:

| Author(s) | Year | Dataset title | Dataset URL | Database and Identifier |
|---|---|---|---|---|
| Shafin K, Pesout T, Lorig-Roach R, Haukness M, Olsen HE, Bosworth C, Armstrong J, Tigyi K, Maurer N, Koren S, Sedlazeck FJ, Marschall T, Mayes S, Costa V, Zook JM, Liu KJ, Kilburn D, Sorensen M, Munson KM, Vollger MR, Monlong J, Garrison E, Eichler EE, Salama S, Haussler D, Green RE, Akeson M, Phillippy A, Miga KH, Carnevali P, Jain M, Paten B | 2020 | Nanopore sequencing and the Shasta toolkit enable efficient de novo assembly of eleven human genomes | https://github.com/human-pangenomics/hpgp-data | Human Pangenome Reference Consortium, hpgp-data |
| Jain M, Koren S, Miga KH, Quick J, Rand AC, Sasani TA, Tyson JR, Beggs AD, Dilthey AT, Fiddes IT, Malla S, Marriott H, Nieto T, O'Grady J, Olsen HE, Pedersen BS, Rhie A, Richardson H, Quinlan AR, Snutch TP, Tee L, Paten B, Phillippy AM, Simpson JT, Loman NJ, Loose M | 2018 | Nanopore sequencing and assembly of a human genome with ultra-long reads | https://github.com/nanopore-wgs-consortium/NA12878/blob/master/Genome.md | Nanopore WGS Consortium, NA12878 |

*Continued on next page*

*Continued*

| Author(s) | Year | Dataset title | Dataset URL | Database and Identifier |
|---|---|---|---|---|
| De Coster W, De Rijk P, De Roeck A, De Pooter T, D'Hert S, Strazisar M, Sleegers K, Van Broeckhoven C | 2019 | Structural variants identified by Oxford Nanopore PromethION sequencing of the human genome | https://www.ebi.ac.uk/ena/browser/view/PRJEB26791 | European Nucleotide Archive, PRJEB26791 |
| Zook JM, Catoe D, McDaniel J, Vang L, Spies N, Sidow A, Weng Z, Liu Y, Mason CE, Alexander N, Henaff E, McIntyre ABR, Chandramohan D, Chen F, Jaeger E, Moshrefi A, Pham K, Stedman W, Liang T, Saghbini M, Dzakula Z, Hastie A, Cao H, Deikus G, Schadt E, Sebra R, Bashir A, Truty RM, Chang CC, Gulbahce N, Zhao K, Ghosh S, Hyland F, Fu Y, Chaisson M, Xiao C, Trow J, Sherry ST, Zaranek AW, Ball M, Bobe J, Estep P, Church GM, Marks P, Kyriazopoulou-Panagiotopoulou S, Zheng GXY, Schnall-Levin M, Ordonez HS, Mudivarti PA, Giorda K, Sheng Y, Rypdal KB, Salit M | 2016 | Extensive sequencing of seven human genomes to characterize benchmark reference materials | ftp://ftp-trace.ncbi.nlm.nih.gov/giab/ftp/ | NCBI Genome in a Bottle FTP, FTP |
| The 1000 Genomes Project Consortium | 2015 | A global reference for human genetic variation | https://www.internationalgenome.org/data-portal/data-collection/30x-grch38 | The International Genome Sample Resource, 30x-grch38 |
| Stunnenberg HG, Abrignani S, Adams D, de Almeida M, Altucci L, Amin V, Amit I, Antonarakis SE, Aparicio S, Arima T, Arrigoni L, Arts R, Asnafi V, Badosa ME, Bae JB, Bassler K, Beck S, Berkman B, Bernstein BE, Hirst M | 2016 | The International Human Epigenome Consortium: A Blueprint for Scientific Collaboration and Discovery | https://www.blueprint-epigenome.eu/ | Blueprint Epigenome, blueprint |
| Bernstein BE, Stamatoyannopoulos JA, Costello JF, Ren B, Milosavljevic A, Meissner A, Kellis M, Marra MA, Beaudet AL, Ecker JR, Farnham PJ, Hirst M, Lander ES, Mikkelsen TS, Thomson JA | 2010 | The NIH Roadmap Epigenomics Mapping Consortium | https://www.ncbi.nlm.nih.gov/geo/roadmap/epigenomics/ | NCBI Gene Expression Omnibus, epigenomics |
| ENCODE Project Consortium | 2012 | An Integrated Encyclopedia of DNA Elements in the Human Genome | https://www.encodeproject.org/ | ENCODE, encodeproject |

*Continued on next page*

*Continued*

| Author(s) | Year | Dataset title | Dataset URL | Database and Identifier |
|---|---|---|---|---|
| Okae H, Chiba H, Hiura H, Hamada H, Sato A, Utsunomiya T, Kikuchi H, Yoshida H, Tanaka A, Suyama M, Arima T | 2014 | Genome-wide analysis of DNA methylation dynamics during early human development | ftp://ftp.ddbj.nig.ac.jp/ddbj_database/dra/fastq/DRA003/DRA003802 | DNA Data Bank of Japan, DRA003802 |
| Steinmetz LM, Hogenesch JB, Kellis M, Batzoglou S, Snyder M | 2013 | Extensive Variation in Chromatin States Across Humans | https://www.ncbi.nlm.nih.gov/sra/?term=SRP030041 | NCBI Sequence Read Archive, SRP030041 |
| Hon GC, Rajagopal N, Shen Y, McCleary DF, Yue F, Dang MD, Ren B | 2013 | Epigenetic memory at embryonic enhancers identified in DNA methylation maps from adult mouse tissues | https://www.ncbi.nlm.nih.gov/geo/query/acc.cgi?acc=GSE42836 | NCBI Gene Expression Omnibus, GSE42836 |
| Tung J, Barreiro LB, Johnson ZP, Hansen KD, Michopoulos V, Toufexis D, Michelini K, Wilson ME, Gilad Y | 2012 | Social environment is associated with gene regulatory variation in the rhesus macaque immune system | https://www.ncbi.nlm.nih.gov/geo/query/acc.cgi?acc=GSE34128 | NCBI Gene Expression Omnibus, GSE34128 |
| Jeong H, Mendizabal I, Berto S, Chatterjee P, Layman T, Usui N, Toriumi K, Douglas C, Singh D, Huh I, Preuss TM, Konopka G, Yi S V | 2021 | Evolution of DNA methylation in the human brain | https://www.ncbi.nlm.nih.gov/geo/query/acc.cgi?acc=GSE151768 | NCBI Gene Expression Omnibus, GSE151768 |
| Dahlet T, Argüeso Lleida A, Al Adhami H, Dumas M, Bender A, Ngondo RP, Tanguy M, Vallet J, Auclair G, Bardet AF, Weber M | 2020 | Genome-wide analysis in the mouse embryo reveals the importance of DNA methylation for transcription integrity | https://www.ncbi.nlm.nih.gov/geo/query/acc.cgi?acc=GSE130735 | NCBI Gene Expression Omnibus, GSE130735 |
| Jung YH, Sauria MEG, Lyu X, Cheema MS, Ausio J, Taylor J, Corces VG | 2017 | Chromatin States in Mouse Sperm Correlate with Embryonic and Adult Regulatory Landscapes | https://www.ncbi.nlm.nih.gov/geo/query/acc.cgi?acc=GSE79226 | NCBI Gene Expression Omnibus, GSE79226 |
| Saenz-de-Juano MD, Ivanova E, Billooye K, Herta A-C, Smitz J, Kelsey G, Anckaert E | 2019 | Genome-wide assessment of DNA methylation in mouse oocytes reveals effects associated with in vitro growth, superovulation, and sexual maturity | https://www.ncbi.nlm.nih.gov/geo/query/acc.cgi?acc=GSE128656 | NCBI Gene Expression Omnibus, GSE128656 |
| Gao F, Niu Y, Sun YE, Lu H, Chen Y, Li S, Kang Y, Luo Y, Si C, Yu J, Li C, Sun N, Si W, Wang H, Ji W, Tan T | 2017 | De novo DNA methylation during monkey pre-implantation embryogenesis | https://www.ncbi.nlm.nih.gov/geo/query/acc.cgi?acc=GSE60166 | NCBI Gene Expression Omnibus, GSE60166 |

*Continued*

| Author(s) | Year | Dataset title | Dataset URL | Database and Identifier |
|---|---|---|---|---|
| Florian Z, Droplaug NM, Olafur TM, Nicolas JW, Tiffany JM, Asgeir S, Gisli HH, Sigurjon AG, Pall M, Helga I, Snædis K, Kristjan, Kristjan FA, Anna H, Julius G, Thorunn R, Ingileif J, Hilma H, Gudmundur IE, Olof S, Isleifur O, Gisli M, Daniel FG, Unnur T, Bjarni VH, Simon NS, Kari S | 2018 | Methylation and expression data for whole-genome human imprinting study | https://doi.org/10.6084/m9.figshare.6816917 | figshare, 10.6084/m9.figshare.6816917 |
| Imbeault M, Helleboid P, Trono D | 2017 | ChIP-exo of human KRAB-ZNFs transduced in HEK 293T cells and KAP1 in hES H1 cells | https://www.ncbi.nlm.nih.gov/geo/query/acc.cgi?acc=GSE78099 | NCBI Gene Expression Omnibus, GSE78099 |
| Takahashi N, Coluccio A, Thorball C. W, Planet E, Shi H, Offner S, Turelli P, Imbeault M, Ferguson-Smith A. C, Trono D | 2019 | ZNF445 is a primary regulator of genomic imprinting | https://www.ncbi.nlm.nih.gov/geo/query/acc.cgi?acc=GSE115387 | NCBI Gene Expression Omnibus, GSE115387 |

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

## Appendix 1

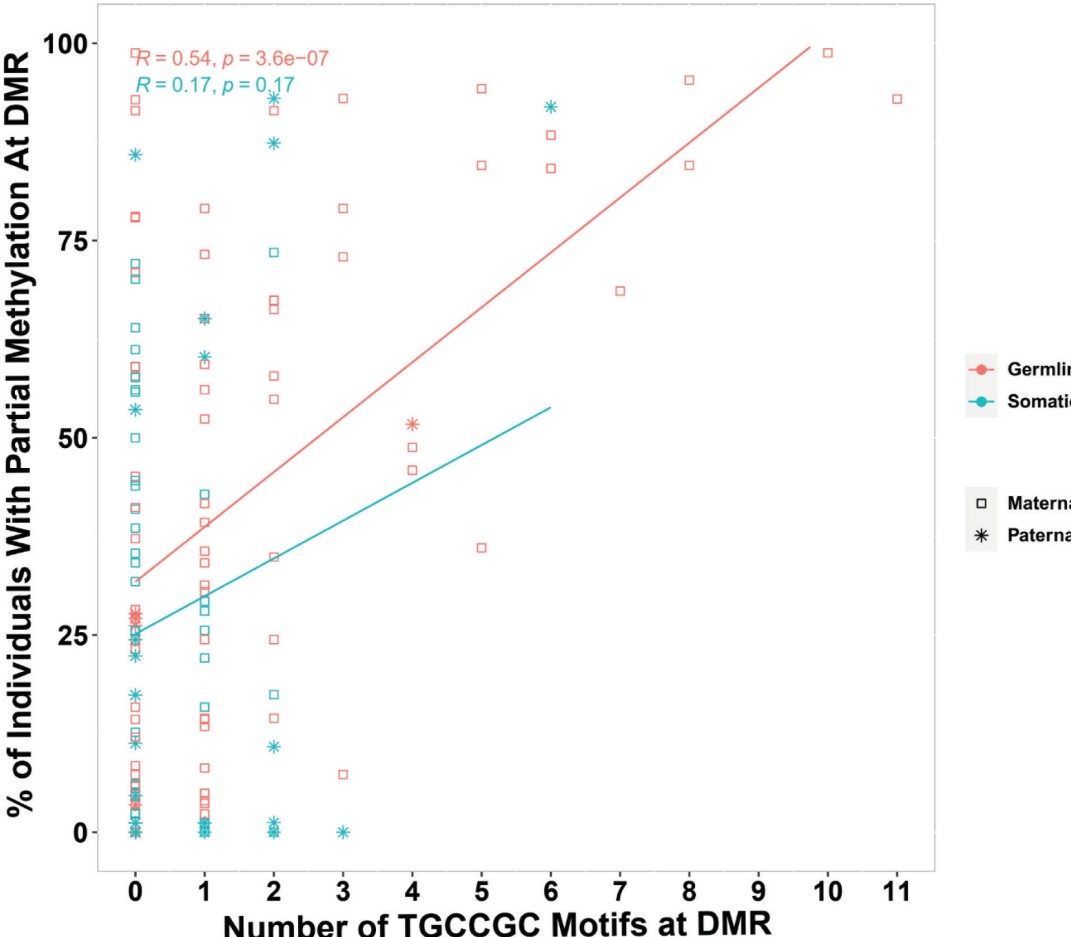

**Appendix 1—figure 1.** Pearson correlation for the number of *ZFP57*-binding motif (TGCCGC) at differentially methylated regions (DMRs) and percent of individuals that demonstrated partial methylation in their whole-genome bisulfite sequencing data at the DMRs.

The online version of this article includes the following source data for appendix 1—figure 1:

• **Appendix 1—figure 1—source data 1.** Number of ZFP57-binding motif (TGCCGC) at differentially methylated regions (DMRs).

## Appendix 2

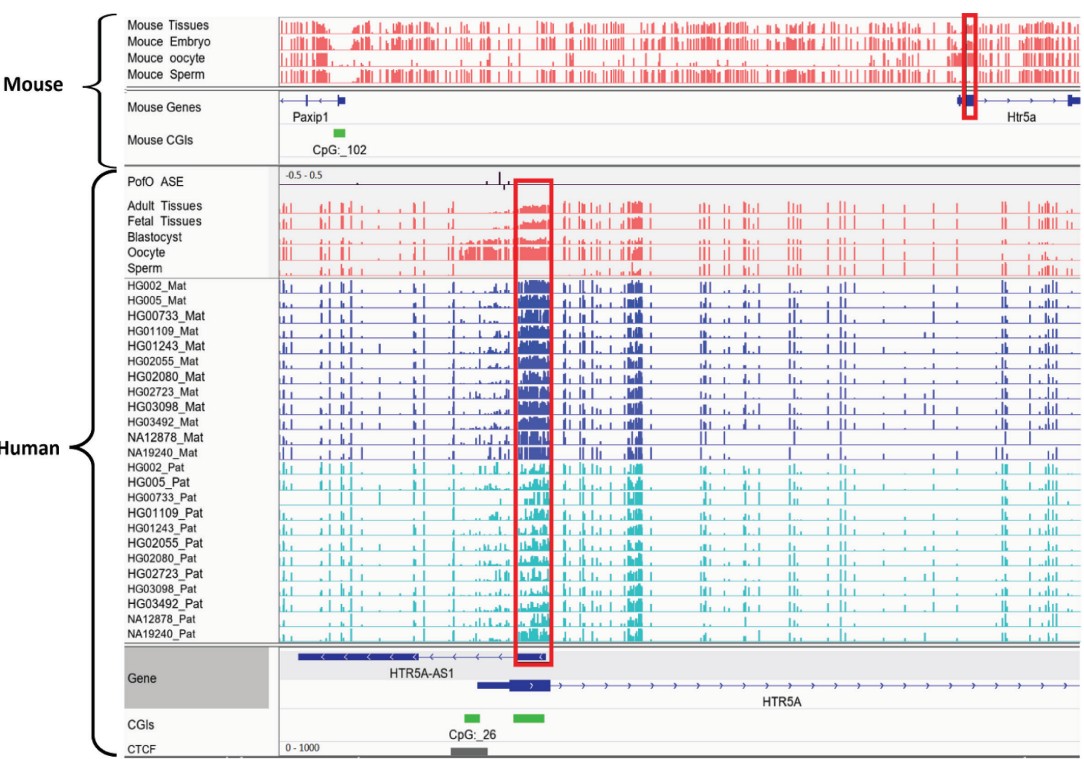

**Appendix 2—figure 1.** Conserved differentially methylated region (DMR) at HTR5A in human and mouse. The known DMR at HTR5A reported to be not conserved in mouse by Court et al. (PMID: 24402520; see supplementary figure S6 from **Court et al., 2014**), however we detected it as conserved due to a different orthologous examined in our study. Court et al. examined CGI 102 which is also not imprinted in our analysis, however the ortholog we examined spans beginning of the HTR5A and is partially methylated which suggests the region is imprinted. Red boxes are showing germline DMRs. The range for all methylation tracks is 0–1. In parent-of-origin (PofO) ASE track, positive or upward bars represent paternal expression bias and negative or downward bars represent maternal expression bias.

## Appendix 3

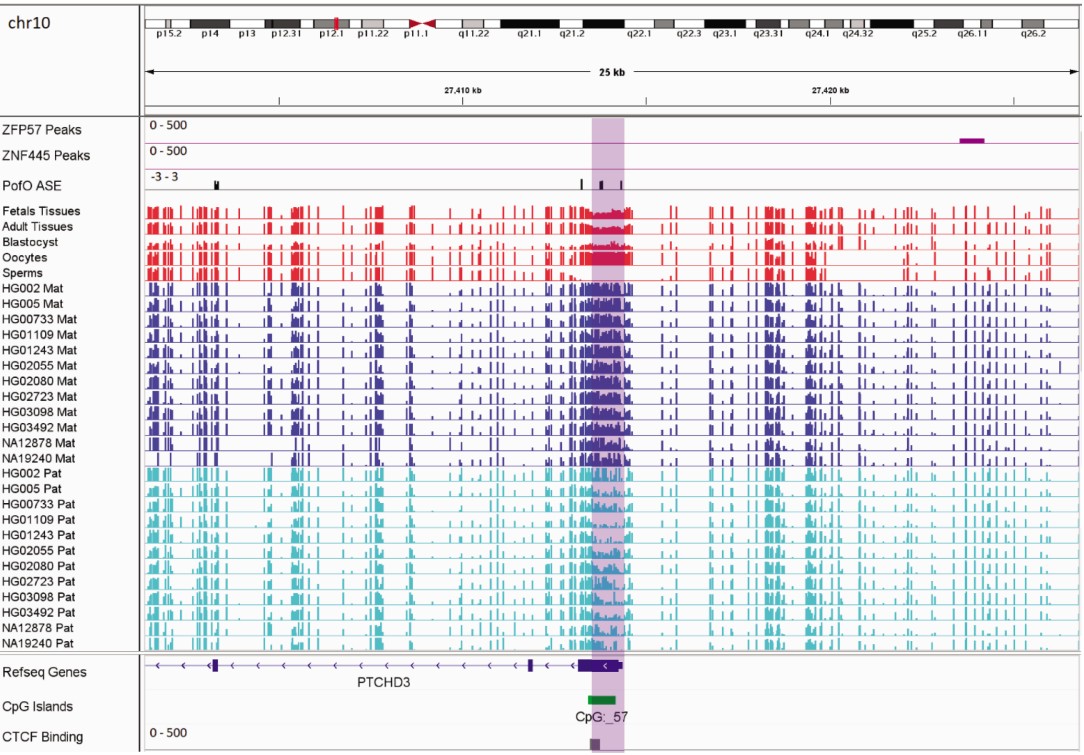

**Appendix 3—figure 1.** Germline maternally methylated differentially methylated region (DMR) in the promoter of paternally expressed PTCHD3 gene. The range for all methylation tracks is 0–1. In parent-of-origin (PofO) ASE track, positive or upward bars represent paternal expression bias and negative or downward bars represent maternal expression bias.

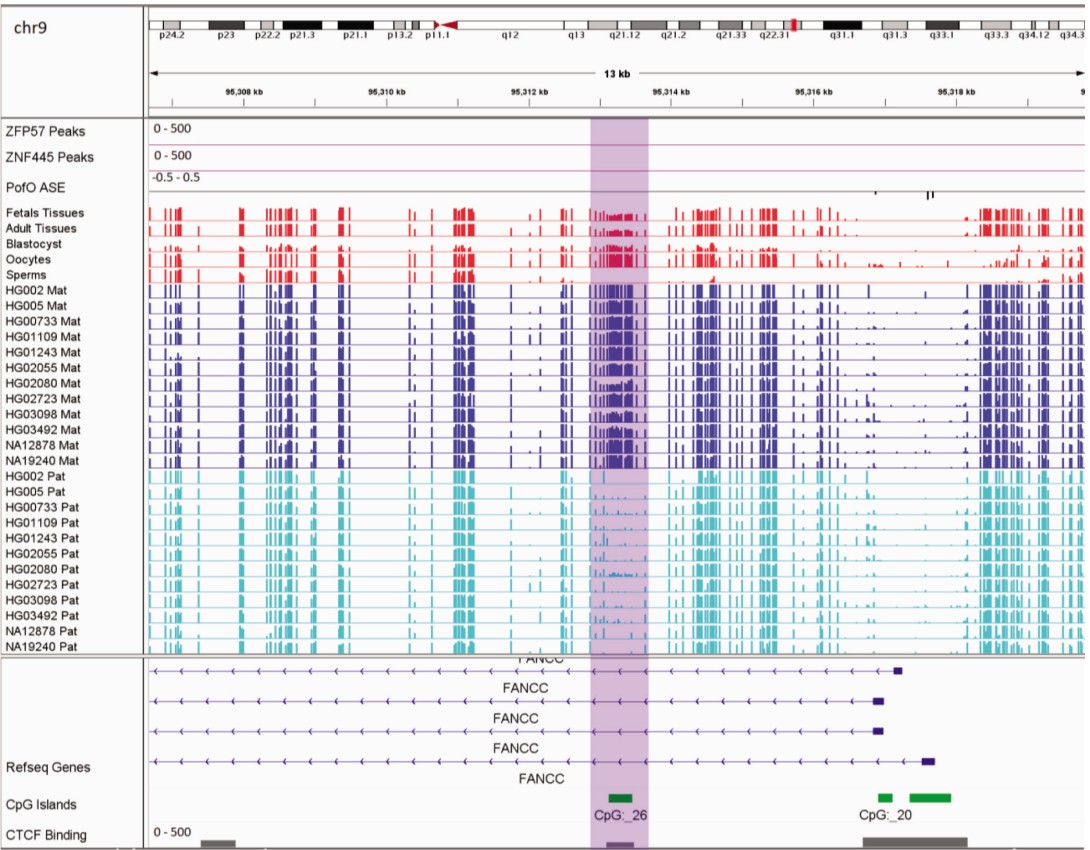

**Appendix 3—figure 2.** Germline maternally methylated differentially methylated region (DMR) in the intron 1 of maternally expressed FANCC gene. The range for all methylation tracks is 0–1. In parent-of-origin (PofO) ASE track, positive or upward bars represent paternal expression bias and negative or downward bars represent maternal expression bias.

