## [Editor Report]

This work uses nanowire sequencing to detect genome-Wide imprinted differentially methylated regions. It will be of broad interest to DNA methylation researchers.

---

## [Decision Letter]

**Decision letter after peer review:**

[Editors’ note: the authors submitted for reconsideration following the decision after peer review. What follows is the decision letter after the first round of review.]

Thank you for submitting the paper "Genome-Wide Detection of Imprinted Differentially Methylated Regions Using Nanopore Sequencing" for consideration by *eLife*. Your article has been reviewed by 3 peer reviewers, and the evaluation has been overseen by a Reviewing Editor and a Senior Editor, with a fair bit of internal discussion among all reviewing parties. The following individual involved in review of your submission has agreed to reveal their identity: Gavin Kelsey (Reviewer #1).

We appreciate that you have put a lot of effort into an important long-standing question and of specific interest here is the application of a newer technology, nanopore sequencing, to that problem. The reviewers agree that your study provides a new perspective with good quality data (exceptions where noted). However, while the report confirms the power of long-read sequencing technology, there is also a general sense that the current manuscript is lacking in biological novelty and the new DMRs would benefit from further investigation. One reviewer suggested examining the DMRs in relation to the possibility of polymorphic imprinting. Alternatively, if the technical advantage including costs / ease of use could be strengthened per comment of another reviewer. As the manuscript stands currently, neither of these two potential selling points are fully utilized. Thus, we cannot offer to publish your manuscript as it stands. We hope that you and your colleagues will benefit from the excellent comments of the reviewers appended to this letter for resubmission elsewhere, or if substantial progress can be made with new experiments, we could potentially reconsider your manuscript with the above comments in mind.

*Reviewer #1:*

The study builds upon a recent report from the authors on the application of long-read nanopore sequencing to the detection of parent-of-origin (PofO) allelic methylation in the human genome (Akbari et al. 2021), with the potential for identification of potentially novel imprinted genes. Nanopore sequencing has previously been shown to be effective in imprinted DMR identification in hybrid mouse crosses (e.g., Gigante et al. 2019). Long-read sequencing has the potential to provide for methylation calls at CpGs to be phased over extended distances with genetic haplotypes and PofO if parent-offspring trios or pedigrees are available. Here, the authors use as discovery datasets 12 LCLs from the 1K Genomes Project and other projects. Candidate PofO DMRs are then parsed through various filters, including methylation level in conventional short-read whole-genome bisulphite sequencing (WGBS) datasets from human gametes, preimplantation embryos as well as somatic tissues, in accordance with the predicted properties of germline DMRs and somatic (or secondary) DMRs. In addition, DMR predictions are compared with well-characterised germline or somatic DMRs (as a test of sensitivity), as well as predictions from a number of other studies that have deployed WGBS or array-based assays of various informative samples (Court et al. 2014, Joshi et al. 2016, Hernandez Mora et al. 2018; Zink et al. 2018). Finally, the authors seek conservation of DMR status in chimpanzee, rhesus macaque, and mouse. This experimental design and implementation look well thought-out and robust.

There have, of course, been numerous attempts to derive a comprehensive list of imprinted loci in the human genome over the last couple of decades. The recent large-scale WGoxBS study of the Icelandic population (Zink et al. 2018) also applied nanopore sequencing, but only in a limited capacity to validate DMRs identified from WGoxBS rather than as a discovery tool as in the current study. As well as demonstrating the power of nanopore sequencing, I think it is important that the current study shows something new, either in relation to undiscovered imprinted loci with potential importance in human development, physiology or disease, or new concepts in imprinting regulation. With some limited further analysis, this level of novelty could be achieved.

The authors report the detection of 76 allelic DMRs not overlapping with previously reported DMRs, of which 28 were determined to be high confidence novel DMRs based on methylation status in the WGBS datasets interrogated. They were able to assign 12 as gDMRs, all but one maternally methylated consistent with the predominance of oocyte methylation of gDMRs. One of the criteria for gDMR status is that the DMR should have "more than 40% methylation in oocyte and less than 20% in sperm and vice versa with methylation difference > 0.25 (lines 188-189)". A threshold of only 40% methylation in one gamete looks rather low for gDMR status. What does 40% methylation mean in molecular terms: that in a population of oocytes there are fully methylated and fully unmethylated alleles; or is there mosaic methylation across the gDMR? From inspection of the heatmap in Figure 3, it seems that very few novel 'gDMRs' have such low methylation in oocytes, so it would be valid to increase the minimum level of methylation required in oocytes to >70%, which would better meet with accepted norms within the imprinting community.

A key determinant of a gDMR is binding of ZFP57 and/or ZNF455 (e.g., Takahashi et al. Genes Dev. 2019 PMID: 30602440), which ensure methylation maintenance especially during methylation reprogramming in preimplantation embryos. The binding motif for ZFP57 at least is known, so the authors should be able to determine whether this motif is present within their candidate gDMRs.

What is striking from the screenshots of many of the novel DMRs (Figure 5 and Suppl. Figures S2, S4, S5, S12, S13, S14, S16) is that they appear to be 'polymorphic', i.e., PofO differential methylation is present in some but not all of the 12 LCLs. I think this needs to be explored in some greater depth. Polymorphic imprinting has been discussed almost since the discovery of imprinting (as in the WT1 locus), but still remains poorly characterised and understood. Recent examples include VTRNA2 (Zink et al. 2018, as well as Silver et al. Genome Biol. 2015 PMID: 26062908). Does the inter-individual variation reflect variation in sequences required for gametic methylation maintenance? For example, is there variation within sequence motifs for ZFP57 binding (if present within the DMR)? This information should be available to the authors. An alternative possibility is that it could reflect variation in establishment of the methylation state in one or other gamete; this cannot be directly addressed but might be a useful speculation to add.

The identification of extended regions of PofO methylation bias in some known imprinted domains (the authors observe this at seven loci) is important and interesting, and extends findings of Zink et al. on the chromosome 15 PWS/AS domain. The authors point to cases in which there is maternal methylation of a promoter-associated gDMR and paternal methylation bias over the expressed gene body. It is possible to offer mechanistic explanation for this, which the authors attempt, speculating that (lines 315-318): "the subtle parental methylation bias is used by cells to express important genes (genes which can regulate other genes in the cluster or have regulatory roles) in an imprinted cluster with higher fidelity through its gene body methylation on active allele." But the authors should consider other explanations for which precedents exist. Thus, the effect may relate to the presence of allelic histone modification bias over the gene body, given that some histone modifications will tend to occupy mutually exclusive locations to DNA methylation; e.g., the transcribed allele is likely to be enriched in H3K36me3 (which attracts de novo DNA methylation), while the non-expressed allele may be enriched in the mutually exclusive modification H3K27me3. For example, at the Kcnq1ot1 and Airn/Igf2r imprinted domains in mouse placental trophoblast, allelic H3K27me3 can span several Megabases (Schertzer et al. Mol Cell 2019 PMID: 31256989; Hanna et al. Genome Biol. 2019 PMID: 31665063; Andergassen et al. PLoS Genet. 2019 PMID: 31329595). There may be ENCODE ChIP-seq datasets that could be informative here. Alternatively, allelic histone modification and consequential DNA methylation bias may be curtailed in regions with sense:antisense transcription on the opposing alleles.

1. More analysis of the possibility of polymorphic imprinting of newly discovered gDMRs.

2. Analysis of ZFP57 binding site motifs in new DMRs: they would be expected to be present in gDMRs with robust maintenance.

3. Apply a more stringent threshold than >40% methylation in oocytes/sperm to call a gDMR.

4. Consideration in the Discussion of mechanisms that could account for extended PofO methylation bias along the lines suggested above.

5. A clear summary table/scheme on DMR status, including: germline or somatic; proximity to known imprinted cluster; consistency/variability of DMR status in the LCLs. The idiogram in Figure 2C has this information but not in a way that it is easy to extract the numbers.

6. The Discussion is on the long side – it would also benefit from some English language editing, particularly in the latter parts.

Other corrections:

Lines 39-40 (and the following sentences): "ICRs are classified as germline (or primary) or somatic (or secondary), hereinafter referred to as gDMR and sDMR." This definition is incorrect. ICRs (imprinting control regions) are by definition germline DMRs in that they are required and sufficient for imprinting of clusters of imprinted genes and sDMRs depend on the existence of the ICR which corresponds to a gDMR.

Lines 50-51: "Loss of imprinting is also widely observed in human cancers." This is not strictly true, as it has been shown that apparent imprinted disruption in tumours is more often a consequence of copy number variation rather than loss of imprinting per se, see Martin-Trujillo et al. 2017 PMID: 28883545.

Line 56: reference given numerically [14-16] rather than by author name.

*Reviewer #2:*

The study is comprehensive in making use of multiple publicly available datasets. It also provides new evolutionary insight into the conservation of DMRs, as it compares data between different mammalian species. As the study discriminates between germline as well as somatic imprints, it therefore provides useful data for the imprinting community to study questions between primary and secondary imprinting marks within imprinted gene clusters. Overall, the study is a comprehensive imprinting resource and Nanopore technical paper, even if it does not seem to provide major conceptual advances over previous studies. The study appears generally well performed and the conclusions are backed up by the data, except in the Discussion section (see below), which could be streamlined and shortened to focus on the main conclusions from this paper. Some improvements could be made in the presentation, when data are only provided in supplementary data table form and not presented in the figures as outlined below.

General issues:

1. The number of acronyms used in this article is very high. To make the article more accessible to a general readership audience, I would recommend reducing their use if possible, or alternatively providing a glossary.

2. The Discussion section seems overly long and speculative and could be substantially shortened, without impacting the overall message of the study.

Specific issues:

1. Figure 1a: "WGBS validation and byond" – should say beyond

2. Line 102-103: "All DMRs which overlapped with previously-reported DMRs displayed consistent PofO with those studies."

Where is this shown? If not, it would be good to show this somewhere.

3. Figure 3b: In order to more clearly identify sDMRs, which are either paternally or maternally methylated, it would be helpful to also add a panel comparing maternal with paternal LCLs.

4. Line 152: "The H3K4me3 histone mark is protective to DNA methylation." This sentence could be easily misinterpreted. It could be rephrased to "The H3K4me3 histone mark is protective against DNA methylation."

5. Section on H3K4me3 and DNA-methylation (line 151-167): A figure panel depicting the relationship between H3K4me3 and DNA-methylation would be helpful instead of only referring to the supplementary data tables.

6. Line 174-175: "Orthologs of the 77/107 detected DMRs showed significant partial methylated in at least one of the three mammals." should be rephrased to something like: "77 of the 107 detected orthologous DMRs showed significant partial methylation in at least one of the three mammals."

*Reviewer #3:*

Akbari, et al. demonstrate the utility of nanopore sequencing data for identification of imprinted genes and differentially methylated regions (DMRs) in lymphocyte cell line (LCL) genomes sequenced by the HPRC and others. Their analysis replicates a large number of imprinted genes and DMRs (96/172) identified by short-read sequencing of bisulfite-converted genomic DNA, but also identifies 76 of novel DMRs and genes in LCLs missed in prior studies. Conversely, a large number (283) of imprinted DMRs missed some of statistical cutoffs in the nanopore sequencing data, although >90% of well-characterized loci do meet these cutoffs. The authors then distinguish somatic from germline imprints found in published gametic and blastocyst DNA methylation datasets, and query corresponding non-human primate and rodent datasets to assess the evolutionary conservation of their novel imprints. Finally, they demonstrate concordant allelic representation of H3K4me3 and mRNA-seq reads of their imprinted genes.

Among the strengths of this study are the comparative analyses to imprinting catalogues from other datasets, and the integration of allelic chromatin marks and gene expression with their DNA methylation analysis.

One drawback of the study in its present form is that it does not address why a relatively large number of imprinted DMRs were not replicated across studies. Cursory analysis of supplementary tables suggests that many of the prior DMRs missed in the nanopore data do demonstrate expected parent-of-origin bias in DNA methylation, but were not picked up in multiple samples likely due to the chosen cutoffs (-/+ 25% methylation difference in at least 4 samples and p < 0.001). As there is no explanation for the chosen nanopore vs. oxBS-seq cutoffs, the reader is left with the impression that the concordance with prior studies is lower than it really is. Conversely, many of the novel DMRs fall near known imprinted DMRs. Among the possible explanations for the large fraction (44%) novel DMRs and missed known DMRs, the authors focus on technical differences, but only briefly mention that some DMRs are cell-type-specific, which may explain why DMRs identified in LCLs may not match DMRs identified in peripheral blood monocytes better, or miss arbitrary statistical cutoffs. One way to address this issue would be to use more sophisticated methods to combine p-values across studies to quantify concordance of imprinted DMRs without applying arbitrary "hard" cutoffs (that don't match between nanopore and oxBS-seq anyway). This is a relatively major point, because while truly novel imprinted DMRs that can only be identified by nanopore are relatively rare, the take-home message of this study could be that while there is very good concordance across methods and most novel (possibly cell-type specific) DMRs fall near known DMRs, there are additional benefits to DNA methylation profiling by nanopore-sequencing.

A second limitation is that the study attributes these differences to long vs. short read technologies without assessing concordance between nanopore and another long-read technology. Because many of the HPRC samples include Pacbio data, one possible way to address this would be to include imprinted DMR analysis in the corresponding Pacbio samples in HPRC.

Overall, the study clearly demonstrates that nanopore-sequencing performs well in profiling DNA methylation, and can identify novel yet conserved imprints, but doesn't highlight its strengths (long-range, single-molecule phased DNA methylation patterns) or its possible weaknesses (single-nucleotide resolution).

I think the authors should consider not applying hard cutoffs in their analysis and instead combine p-values or use rank-based metrics to assess concordance/novelty as suggested above. The concordance appears better than described by the authors, and in general, I feel the authors don't need to draw as many distinctions to the prior work to "sell" the reader on nanopore sequencing.

There are other benefits they could use to draw these points of distinction however, for example by looking at concordance of methylation patterns across single nanopore molecules, or including Pacbio reads and presenting a more informative cost/benefit analysis to guide readers in choosing DNA methylation profiling approaches (oxBS-seq, nanopore, Pacbio). See: "a cheap and easy way to call ICRs".

Apart from the important cutoff-agnostic analysis, and possibly including Pacbio samples, the manuscript could be improved mainly in tone and attention to detail. I noticed some sentence structure errors, but more importantly felt some sentences served to unnecessarily "sell" this study, either by sounding too critical with prior studies or or providing little support, e.g. "cheap and easy" above, or "individuals more representative of the human population" (with only 12 samples). The authors could also re-assess the first sentence of their discussion, and should maybe cite: https://pubmed.ncbi.nlm.nih.gov/33230324/

I think these could be improved without taking anything away from what is an interesting and informative study in its own right. *eLife* would be a good fit for this study, which could be accepted with relatively minor revisions and only limited added analysis.

[Editors’ note: further revisions were suggested prior to acceptance, as described below.]

Thank you for resubmitting your work entitled "Genome-Wide Detection of Imprinted Differentially Methylated Regions Using Nanopore Sequencing" for further consideration by *eLife*. Your revised article has been evaluated by Jessica Tyler (Senior Editor) and a Reviewing Editor.

The manuscript has been improved but there are some remaining issues that need to be addressed, as outlined below:

Everyone felt that the manuscript has been significantly improved. However, the following issues remain:

1. The processed annotated tracks must be deposited in a data repository and released at the time of publication.

2. Comment on the high incidence of polymorphic imprinting in the discussion.

3. Test for sequence variation in ZFP57 binding sites in polymorphically methylated gDMRs if sequence information is available for the LCLs and other informative samples for which DNA methylation data exist.

4. More rigor is required in the analysis of allelic enrichment of H3K36me3 and H3K27me3 at domains of extended PofO DNA methylation bias compared with those imprinted regions that do not, and compute the allelic read scores and provide an appropriate summary plot.

More details on these points are given below, as well as minor points that need to be addressed also.

*Reviewer #2 (Recommendations for the authors):*

The authors addressed some (not all) of the concerns raised in the first round of review, often differently than suggested (which is acceptable).

The data availability however, is not acceptable in my view; given that the bioinformatic approach to DNA methylation phasing is computationally sophisticated and was published elsewhere already (Akbari, et al., 2021), it should be expected that the processed nanometh tracks (shown in Figure 3c, 6-8 and the extensive supplement) are made available via a data repository (e.g. GEO or Mendeley data). Inclusion of processed data tracks is obligatory for GEO submissions, and recreating these data tracks from the raw data places an undue burden on labs wishing to query these data for their loci of interest. These data tracks should include all chromosomes, including sex chromosomes, to enable others to re-run statistical analyses with their own (p-value and DNA methyation change) cutoffs.

*Reviewer #3 (Recommendations for the authors):*

In my review of the original manuscript, I suggested that it was important that the study showed something new, either in relation to undiscovered imprinted loci with potential importance in human development, physiology or disease, or new concepts in imprinting regulation. Specifically, I recommended further analysis of:

– polymorphic imprinting

– analysis of ZFP57/ZNF445 binding sites within candidate gDMRs

– stricter threshold for methylation levels in gametes to call candidate gDMRs

and additional improvements including:

– more focussed discussion

– clear summary table

In general, these points have been addressed well in the revised manuscript.

"Novel Imprinted DMRs Display Inter-Individual Variation"

I think this is an important addition to the manuscript that demonstrates the majority of the novel DMRs identified in the current study could represent 'polymorphic' imprinting and they are not consistently methylated in blood samples from 87 individuals, in comparison to 'well-characterised' DMRs. Some of the novel DMRs exhibited partial methylation consistent with imprinted status in as few as 1-2% of individuals, which could explain why they had not been detected in previous studies. The relevant analysis is performed well.

On the other hand, it is an omission that the authors do not comment on this high incidence of polymorphic imprinting in the discussion. The authors do need to return to this finding in their discussion. Although at this point, they are not able to provide much speculation for why these loci exhibit inter-individual variation, there are multiple implications of the finding.

Regarding the opening line of this section, "Imprinted methylation can display variation across individuals due to environmental and genetic factors", I think this statement could be modified, as generally speaking imprinting (i.e., well-characterised imprints) is consistent between individuals and resistant to environmental factors (with procedures associated with assisted reproduction techniques possibly the most likely 'environment' to lead to instability of methylation at gDMRs). Therefore, rephrase to something like: "Although imprinted methylation is generally regarded is consistent between individuals and resistant to environmental factors, there are examples of polymorphic imprinting……."

"Determination of Germline versus Somatic Status of Novel Imprinted DMRs"

This section now includes the analysis of ZFP57/ZNF445 binding data – published ChIP-seq datasets from hESCs and HEK293 cells. Interestingly, 44% of the novel gDMRs have binding for one or both of these proteins in these cells, compared with 49% of characterised gDMRs. A difference in binding by these factors does not therefore appear to relate to the high incidence of polymorphic imprinting amongst novel DMRs, but the possibility that there is sequence variation in the ZFP57 binding sites between the sequenced samples has not been addressed.

It is also welcomed that the authors apply the more stringent thresholds for gDMR classification in this section.

"Enriched Allelic H3K36me3 and H3K27me3 Histone Marks at Contiguous Blocks"

It is interesting to see the differential allelic enrichment for these two histone modifications across the seven extended domains that display biased parent-of-origin DNA methylation. The authors draw a distinction with other imprinted domains without this extended PofO bias. This conclusion is based entirely on inspection of screenshots of ChIP-seq data, examples of which are provided in Figure 8 and Supplementary Figures 14-26. I think the authors could be a little more rigorous and compute the allelic read scores and provide an appropriate summary plot.

1. It is an omission that the authors do not comment on this high incidence of polymorphic imprinting in the discussion. The authors do need to return to this finding in their discussion.

2. Test for sequence variation in ZFP57 binding sites in polymorphically methylated gDMRs if sequence information is available for the LCLs and other informative samples for which DNA methylation data exist.

3. The authors should be more rigorous in their analysis of allelic enrichment of H3K36me3 and H3K27me3 at domains of extended PofO DNA methylation bias compared with those imprinted regions that do not, and compute the allelic read scores and provide an appropriate summary plot.

4. Rephrase statement: "Imprinted methylation can display variation across individuals due to environmental and genetic factors".

---

## [Author Response]

[Editors’ note: the authors resubmitted a revised version of the paper for consideration. What follows is the authors’ response to the first round of review.]

Thank you for considering our manuscript entitled "Genome-Wide Detection of Imprinted Differentially Methylated Regions Using Nanopore Sequencing". We greatly appreciate the reviewers’ constructive comments and positive evaluation of the presented work. We performed a new set of analyses to address the comments regarding our work. We believe the new experiments address the concerns raised by the reviewers and we would like to resubmit our manuscript for publication within eLife journal. We summarize the main features of our revision which addresses all of the reviewers’ concerns:

We set a more relaxed threshold to detect differentially methylated regions (DMRs) within the nanopore data.We added more WGBS tissue and blood samples to further improve the confirmation of the detected DMRs.We have now included WGBS data from 87 individuals and examined the evidence for polymorphic imprinting.For the detected germline DMRs we set a more stringent threshold of 70%.H3K36me3 and H3K27me3 ChIP-seq data are now included to examine allelic status of these histone marks in the detected imprinted blocks.We used ChIP-seq data for ZFP57 and ZNF445 to determine if the germline DMRs are protected using these proteins during the second reprogramming step of human development.We also edited the manuscript and addressed requested changes and modifications in the figures and the manuscript body by the reviewers.

Reviewer #1:The study builds upon a recent report from the authors on the application of long-read nanopore sequencing to the detection of parent-of-origin (PofO) allelic methylation in the human genome (Akbari et al. 2021), with the potential for identification of potentially novel imprinted genes. Nanopore sequencing has previously been shown to be effective in imprinted DMR identification in hybrid mouse crosses (e.g., Gigante et al. 2019). Long-read sequencing has the potential to provide for methylation calls at CpGs to be phased over extended distances with genetic haplotypes and PofO if parent-offspring trios or pedigrees are available. Here, the authors use as discovery datasets 12 LCLs from the 1K Genomes Project and other projects. Candidate PofO DMRs are then parsed through various filters, including methylation level in conventional short-read whole-genome bisulphite sequencing (WGBS) datasets from human gametes, preimplantation embryos as well as somatic tissues, in accordance with the predicted properties of germline DMRs and somatic (or secondary) DMRs. In addition, DMR predictions are compared with well-characterised germline or somatic DMRs (as a test of sensitivity), as well as predictions from a number of other studies that have deployed WGBS or array-based assays of various informative samples (Court et al. 2014, Joshi et al. 2016, Hernandez Mora et al. 2018; Zink et al. 2018). Finally, the authors seek conservation of DMR status in chimpanzee, rhesus macaque, and mouse. This experimental design and implementation look well thought-out and robust.There have, of course, been numerous attempts to derive a comprehensive list of imprinted loci in the human genome over the last couple of decades. The recent large-scale WGoxBS study of the Icelandic population (Zink et al. 2018) also applied nanopore sequencing, but only in a limited capacity to validate DMRs identified from WGoxBS rather than as a discovery tool as in the current study. As well as demonstrating the power of nanopore sequencing, I think it is important that the current study shows something new, either in relation to undiscovered imprinted loci with potential importance in human development, physiology or disease, or new concepts in imprinting regulation. With some limited further analysis, this level of novelty could be achieved.The authors report the detection of 76 allelic DMRs not overlapping with previously reported DMRs, of which 28 were determined to be high confidence novel DMRs based on methylation status in the WGBS datasets interrogated. They were able to assign 12 as gDMRs, all but one maternally methylated consistent with the predominance of oocyte methylation of gDMRs. One of the criteria for gDMR status is that the DMR should have "more than 40% methylation in oocyte and less than 20% in sperm and vice versa with methylation difference > 0.25 (lines 188-189)". A threshold of only 40% methylation in one gamete looks rather low for gDMR status. What does 40% methylation mean in molecular terms: that in a population of oocytes there are fully methylated and fully unmethylated alleles; or is there mosaic methylation across the gDMR? From inspection of the heatmap in Figure 3, it seems that very few novel 'gDMRs' have such low methylation in oocytes, so it would be valid to increase the minimum level of methylation required in oocytes to >70%, which would better meet with accepted norms within the imprinting community.A key determinant of a gDMR is binding of ZFP57 and/or ZNF455 (e.g., Takahashi et al. Genes Dev. 2019 PMID: 30602440), which ensure methylation maintenance especially during methylation reprogramming in preimplantation embryos. The binding motif for ZFP57 at least is known, so the authors should be able to determine whether this motif is present within their candidate gDMRs.What is striking from the screenshots of many of the novel DMRs (Figure 5 and Suppl. Figures S2, S4, S5, S12, S13, S14, S16) is that they appear to be 'polymorphic', i.e., PofO differential methylation is present in some but not all of the 12 LCLs. I think this needs to be explored in some greater depth. Polymorphic imprinting has been discussed almost since the discovery of imprinting (as in the WT1 locus), but still remains poorly characterised and understood. Recent examples include VTRNA2 (Zink et al. 2018, as well as Silver et al. Genome Biol. 2015 PMID: 26062908). Does the inter-individual variation reflect variation in sequences required for gametic methylation maintenance? For example, is there variation within sequence motifs for ZFP57 binding (if present within the DMR)? This information should be available to the authors. An alternative possibility is that it could reflect variation in establishment of the methylation state in one or other gamete; this cannot be directly addressed but might be a useful speculation to add.The identification of extended regions of PofO methylation bias in some known imprinted domains (the authors observe this at seven loci) is important and interesting, and extends findings of Zink et al. on the chromosome 15 PWS/AS domain. The authors point to cases in which there is maternal methylation of a promoter-associated gDMR and paternal methylation bias over the expressed gene body. It is possible to offer mechanistic explanation for this, which the authors attempt, speculating that (lines 315-318): "the subtle parental methylation bias is used by cells to express important genes (genes which can regulate other genes in the cluster or have regulatory roles) in an imprinted cluster with higher fidelity through its gene body methylation on active allele." But the authors should consider other explanations for which precedents exist. Thus, the effect may relate to the presence of allelic histone modification bias over the gene body, given that some histone modifications will tend to occupy mutually exclusive locations to DNA methylation; e.g., the transcribed allele is likely to be enriched in H3K36me3 (which attracts de novo DNA methylation), while the non-expressed allele may be enriched in the mutually exclusive modification H3K27me3. For example, at the Kcnq1ot1 and Airn/Igf2r imprinted domains in mouse placental trophoblast, allelic H3K27me3 can span several Megabases (Schertzer et al. Mol Cell 2019 PMID: 31256989; Hanna et al. Genome Biol. 2019 PMID: 31665063; Andergassen et al. PLoS Genet. 2019 PMID: 31329595). There may be ENCODE ChIP-seq datasets that could be informative here. Alternatively, allelic histone modification and consequential DNA methylation bias may be curtailed in regions with sense:antisense transcription on the opposing alleles.1. More analysis of the possibility of polymorphic imprinting of newly discovered gDMRs.

We have addressed this using 119 blood samples from 87 individuals. The detailed analysis and results are in the new section of the manuscript entitled “Novel Imprinted DMRs Display Inter-Individual Variation”. Relying on the fact that imprinted regions display partial methylation of around 50%, we calculated the proportion of individuals displaying partial methylation. We accurately confirmed DMRs that have already been reported to be polymorphic or not. For example, *VTRNA2-1, IGF2*, *RB1*, *PARD6G*, *CHRNE*, and *IGF2R* are known to be polymorphic ^1,2^. We have detected partial methylation in 2-65% of the individuals in our analysis for these regions (M ± SD = 40% ± 22%; Supplementary Table 5) indication of their interindividual variability. *ZNF331* DMR is known to be consistently imprinted across individuals ^2^ and we also detected partial methylation in 99% of the individuals at this region (Supplementary Table 5).

We performed the same analysis on the novel DMRs. Novel DMRs demonstrated interindividual variation ranging from 1-73% (M ± SD=23.6% ± 19.2%; Table 1). The novel maternal sDMR near *BTBD7P1* is the most consistent with partial methylation in 73% of the individuals. On the other hand, the novel paternal somatic DMRs within *AC092296.3* and *UBAC2* are the most variable with partial methylation in 1% of the individuals (Table 1). However, because of the unavailability of the genomic data from the 87 individuals and the small sample size of 12 cell lines we cannot establish an association of the polymorphic methylation and possible underlying genetic factors.

2. Analysis of ZFP57 binding site motifs in new DMRs: they would be expected to be present in gDMRs with robust maintenance.

We used ChIP-seq data for ZFP57 and ZNF445. 44% of the novel germline DMRs and 49% of the reported DMRs overlapped with the ChIP-seq peak calls for ZFP57 and/or ZNF445 (Figure 3C; Supplementary Table 4). The results have been included in the “Determination of Germline versus Somatic Status of Novel Imprinted DMRs” section of the manuscript.

3. Apply a more stringent threshold than >40% methylation in oocytes/sperm to call a gDMR.

More stringent threshold (>70%) was used to determine germline DMRs. Using this threshold, two of the novel DMRs including *NFE2L3P1* and *ZNF714* that previously detected as germline now tagged as somatic. The results of this analysis are provided in the “Determination of Germline versus Somatic Status of Novel Imprinted DMRs” section of the manuscript.

4. Consideration in the Discussion of mechanisms that could account for extended PofO methylation bias along the lines suggested above.

To address this, we used H3K36me3 and H3K27me3 ChIP-seq data. At the large imprinted blocks, we observed allelic H3K36me3 marking on the hypermethylated and expressed allele and H3K27me3 on the hypomethylated and silenced allele. We did not observe such allelic histone marks on other imprinted clusters without large blocks of PofO methylation bias. These results suggest high expression of active allele and allelic deposition of histone marks at these blocks are involve in the regulation of subtle PofO methylation biases. The detailed results of these analyses are included in the “Enriched Allelic H3K36me3 and H3K27me3 Histone Marks at Contiguous Blocks” section of the manuscript.

5. A clear summary table/scheme on DMR status, including: germline or somatic; proximity to known imprinted cluster; consistency/variability of DMR status in the LCLs. The idiogram in Figure 2C has this information but not in a way that it is easy to extract the numbers.

We added a new table (Table 1) which includes a summary information of the detected DMRs.

6. The Discussion is on the long side – it would also benefit from some English language editing, particularly in the latter parts.

The manuscript has been edited to improve clarity and correctness of language.

Reviewer #2:The study is comprehensive in making use of multiple publicly available datasets. It also provides new evolutionary insight into the conservation of DMRs, as it compares data between different mammalian species. As the study discriminates between germline as well as somatic imprints, it therefore provides useful data for the imprinting community to study questions between primary and secondary imprinting marks within imprinted gene clusters. Overall, the study is a comprehensive imprinting resource and Nanopore technical paper, even if it does not seem to provide major conceptual advances over previous studies. The study appears generally well performed and the conclusions are backed up by the data, except in the Discussion section (see below), which could be streamlined and shortened to focus on the main conclusions from this paper. Some improvements could be made in the presentation, when data are only provided in supplementary data table form and not presented in the figures as outlined below.General issues:1. The number of acronyms used in this article is very high. To make the article more accessible to a general readership audience, I would recommend reducing their use if possible, or alternatively providing a glossary.

We provided a glossary at the end of the manuscript and reduced the frequency of acronyms in the main text as well.

“Glossary

LCLs: Lymphoblastoid cell lines (LCLs) are immortalized cell lines derived from the Blymphocytes in the peripheral blood.

WGBS: Whole-genome bisulfite sequencing (WGBS) is an approach for detection of DNA methylation. In this strategy, DNA is treated with bisulfite to convert cytosine to uracil, with 5mC and 5hmC remaining intact. Sequencing and mapping of this converted DNA allows the detection of 5mC.

DMA and DMR: Differential methylation analysis (DMA) is the process of comparing two samples or, in case of imprinting, two alleles to detect differentially methylated regions (DMRs). DMRs are regions that show methylation difference either between two samples or, in case of imprinting, between the paternal and maternal alleles.

PofO: Parent-of-origin (PofO) refers to the maternal or paternal origin of an allele.

ASE: Allele-specific expression (ASE) refers to the preferential expression of one allele, either maternal or paternal allele.

ICR: Imprinting control region (ICR) is region of usually a few kilobases that determines allelespecific expression of the imprinted genes based upon the parent-of-origin. ICR is usually controlled through DNA methylation and is a DMR in which only paternal or maternal allele is methylated. ICR is a *cis-acting* regulatory region which means it regulate the expression of gene(s) only on the same chromosomal allele.

UPD: Uniparental disomies (UPD). During normal human development, each individual inherit one paternal and one maternal copy of each chromosome. In some abnormalities, both copy of a chromosome, or of part of a chromosome, are coming from one parent, it is called UPD.

CGI: CpG islands (CGIs) are stretch of DNA, usually 500–1500 bp long, with more than 50% GC content and a CG: GC ratio of more than 0.6.”

2. The Discussion section seems overly long and speculative and could be substantially shortened, without impacting the overall message of the study.

Discussion section has been modified accordingly.

Specific issues:1. Figure 1a: "WGBS validation and byond" – should say beyond

Figure has been corrected.

2. Line 102-103: "All DMRs which overlapped with previously-reported DMRs displayed consistent PofO with those studies."Where is this shown? If not, it would be good to show this somewhere.

This information is available in the supplementary table 3. In the manuscript, we have also added the text to refer readers to the supplementary table 3.

“101 overlapped with 103 previously reported DMRs with consistent PofO, while the remaining 99 were novel (Figure 1C; Supplementary Table 3)”

3. Figure 3b: In order to more clearly identify sDMRs, which are either paternally or maternally methylated, it would be helpful to also add a panel comparing maternal with paternal LCLs.

The panel is added to the figure 3b with the maternal methylation of LCLs on the X axis and the paternal methylation on the Y axis.

4. Line 152: "The H3K4me3 histone mark is protective to DNA methylation." This sentence could be easily misinterpreted. It could be rephrased to "The H3K4me3 histone mark is protective against DNA methylation."

The sentence has been modified as suggested by the reviewer.

5. Section on H3K4me3 and DNA-methylation (line 151-167): A figure panel depicting the relationship between H3K4me3 and DNA-methylation would be helpful instead of only referring to the supplementary data tables.

A new figure (Figure 4) has been added to the manuscript to represent the results from the allelic H3K4me3 analysis at each heterozygous single nucleotide variant.

6. Line 174-175: "Orthologs of the 77/107 detected DMRs showed significant partial methylated in at least one of the three mammals." should be rephrased to something like: "77 of the 107 detected orthologous DMRs showed significant partial methylation in at least one of the three mammals."

The manuscript has been edited as suggested by the reviewer.

Reviewer #3:Akbari, et al. demonstrate the utility of nanopore sequencing data for identification of imprinted genes and differentially methylated regions (DMRs) in lymphocyte cell line (LCL) genomes sequenced by the HPRC and others. Their analysis replicates a large number of imprinted genes and DMRs (96/172) identified by short-read sequencing of bisulfite-converted genomic DNA, but also identifies 76 of novel DMRs and genes in LCLs missed in prior studies. Conversely, a large number (283) of imprinted DMRs missed some of statistical cutoffs in the nanopore sequencing data, although >90% of well-characterized loci do meet these cutoffs. The authors then distinguish somatic from germline imprints found in published gametic and blastocyst DNA methylation datasets, and query corresponding non-human primate and rodent datasets to assess the evolutionary conservation of their novel imprints. Finally, they demonstrate concordant allelic representation of H3K4me3 and mRNA-seq reads of their imprinted genes.Among the strengths of this study are the comparative analyses to imprinting catalogues from other datasets, and the integration of allelic chromatin marks and gene expression with their DNA methylation analysis.One drawback of the study in its present form is that it does not address why a relatively large number of imprinted DMRs were not replicated across studies. Cursory analysis of supplementary tables suggests that many of the prior DMRs missed in the nanopore data do demonstrate expected parent-of-origin bias in DNA methylation, but were not picked up in multiple samples likely due to the chosen cutoffs (-/+ 25% methylation difference in at least 4 samples and p < 0.001). As there is no explanation for the chosen nanopore vs. oxBS-seq cutoffs, the reader is left with the impression that the concordance with prior studies is lower than it really is. Conversely, many of the novel DMRs fall near known imprinted DMRs. Among the possible explanations for the large fraction (44%) novel DMRs and missed known DMRs, the authors focus on technical differences, but only briefly mention that some DMRs are cell-type-specific, which may explain why DMRs identified in LCLs may not match DMRs identified in peripheral blood monocytes better, or miss arbitrary statistical cutoffs. One way to address this issue would be to use more sophisticated methods to combine p-values across studies to quantify concordance of imprinted DMRs without applying arbitrary "hard" cutoffs (that don't match between nanopore and oxBS-seq anyway). This is a relatively major point, because while truly novel imprinted DMRs that can only be identified by nanopore are relatively rare, the take-home message of this study could be that while there is very good concordance across methods and most novel (possibly cell-type specific) DMRs fall near known DMRs, there are additional benefits to DNA methylation profiling by nanopore-sequencing.A second limitation is that the study attributes these differences to long vs. short read technologies without assessing concordance between nanopore and another long-read technology. Because many of the HPRC samples include Pacbio data, one possible way to address this would be to include imprinted DMR analysis in the corresponding Pacbio samples in HPRC.Overall, the study clearly demonstrates that nanopore-sequencing performs well in profiling DNA methylation, and can identify novel yet conserved imprints, but doesn't highlight its strengths (long-range, single-molecule phased DNA methylation patterns) or its possible weaknesses (single-nucleotide resolution).I think the authors should consider not applying hard cutoffs in their analysis and instead combine p-values or use rank-based metrics to assess concordance/novelty as suggested above. The concordance appears better than described by the authors, and in general, I feel the authors don't need to draw as many distinctions to the prior work to "sell" the reader on nanopore sequencing.There are other benefits they could use to draw these points of distinction however, for example by looking at concordance of methylation patterns across single nanopore molecules, or including Pacbio reads and presenting a more informative cost/benefit analysis to guide readers in choosing DNA methylation profiling approaches (oxBS-seq, nanopore, Pacbio). See: "a cheap and easy way to call ICRs".Apart from the important cutoff-agnostic analysis, and possibly including Pacbio samples, the manuscript could be improved mainly in tone and attention to detail. I noticed some sentence structure errors, but more importantly felt some sentences served to unnecessarily "sell" this study, either by sounding too critical with prior studies or or providing little support, e.g. "cheap and easy" above, or "individuals more representative of the human population" (with only 12 samples). The authors could also re-assess the first sentence of their discussion, and should maybe cite: https://pubmed.ncbi.nlm.nih.gov/33230324/

We have used more relaxed cut-off for the detection of the DMRs (0.20 as methylation difference cut-off instead of 0.25). This resulted in including 27 more DMRs of which 4 mapped to reported intervals. Because imprinted methylation is variable across individuals and also tissues, DMRs which reported by multiple studies are more likely to be consistent across individuals while other DMRs are likely to be more variable. We have included 119 WGBS datasets from 87 individuals and assessed polymorphic imprinting. As demonstrated in figure 1C, considerable number of imprinted DMRs detected in different studies are not overlapping between studies. Our analysis suggest polymorphic imprinting can explain this non-overlapping DMRs. The DMRs that detected by at least two studies demonstrated more consistency across individuals with average DMR frequency of 41.2% (SD = 33%) while DMRs detected in a single study showed more variability (M ± SD = 10.6% ± 15.4%) (Supplementary Table 5).

In the white paper published by PacBio technology, a 250x coverage per-strand is needed for PacBio to detect methylation reliably ^3^. Therefore, in a mammalian size genome this technology is prohibitively expensive. Moreover, in the current study our goal is to detect imprinted methylation genome-wide using nanopore long-reads that we and others previously showed could detect allelic methylation at even shallow coverage ^4,5^. We do not necessarily want to promote nanopore sequencing but our results indicate that nanopore sequencing represents a modality suitable for this task. We have tried to present a comparison with previous studies that we think can be useful for the readers and the community. However, we have modified sentences that might imply we are trying to overly promote nanopore sequencing.

Regarding the “randomly selected regions” We have modified the analysis and also included 100 random resampling. This has improved the confirmation of DMRs and the accuracy of the approach. The detailed analysis is provided in the “Confirmation of Novel Imprinted DMRs” section in the manuscript.

Regarding the "cheap and easy", Nanopore can phase significantly higher number of CpGs in a single sample compare to short-read. Therefore, in cases where we need large cohort of samples to achieve higher resolution of phased CpGs using short-reads, by using nanopore we can achieve this with mush smaller cohort size. By "cheap and easy" we meant that it will help to reduce the cost by reducing the sample size. However, we have modified the respective sentences in the last paragraph of the Discussion section to not be misleading.

“We also showed that nanopore sequencing has the ability to achieve a higher resolution of phased CpGs using a small sample size and allows for the calling of imprinted methylation in a single sample, potentially reducing the cost by reducing the sample size.”

I think these could be improved without taking anything away from what is an interesting and informative study in its own right. eLife would be a good fit for this study, which could be accepted with relatively minor revisions and only limited added analysis.

We thank the reviewer for this endorsement.

References

Joshi, R. S. *et al.* DNA Methylation Profiling of Uniparental Disomy Subjects Provides a Map of Parental Epigenetic Bias in the Human Genome. *Am. J. Hum. Genet.* 99, 555–566 (2016).

Zink, F. *et al.* Insights into imprinting from parent-of-origin phased methylomes and transcriptomes. *Nat. Genet.* 50, 1542–1552 (2018).

Biosciences, P. Detecting DNA base modifications using single molecule, real-time sequencing. *White Pap. Base Modif.* (2015) doi:https://www.pacb.com/wpcontent/uploads/2015/09/WP_Detecting_DNA_Base_Modifications_Using_SMRT_Seque ncing.pdf.

Gigante, S. *et al.* Using long-read sequencing to detect imprinted DNA methylation. *Nucleic Acids Res.* 47, e46–e46 (2019).

Akbari, V. *et al.* Megabase-scale methylation phasing using nanopore long reads and NanoMethPhase. *Genome Biol.* 22, 68 (2021).

[Editors’ note: what follows is the authors’ response to the second round of review.]

The manuscript has been improved but there are some remaining issues that need to be addressed, as outlined below:Everyone felt that the manuscript has been significantly improved. However, the following issues remain:1. The processed annotated tracks must be deposited in a data repository and released at the time of publication.

We have deposited methylation and histone modification tracks to Mendeley data repository (https://data.mendeley.com/datasets/f4k2gytbh5/1; doi: 10.17632/f4k2gytbh5.1). We have also provided the “Data availability” section:

“Data availability

In this study we have used various nanopore sequencing, WGBS, and ChIP-seq datasets. The source of each dataset is provided under the “Materials and methods” section under the appropriate subsection. We also used PofO allele-specific expression data from Zink et al. (2018; https://doi.org/10.6084/m9.figshare.6816917) to generate ASE track for IGV. Phased DNA methylation tracks for the 12 human LCL samples and other tracks including ASE, histone modifications and WGBS methylation tracks generated in this study are publicly available through the Mendeley data repository (https://data.mendeley.com/datasets/f4k2gytbh5/1; DOI: 10.17632/f4k2gytbh5.1). Codes used for partial methylation analysis of DMRs in WGBS datasets (PartialMethylation_AtDMR.sh) and binomial test analysis of histone modifications

(CountReadsAtSNV.py and Binomial_test.py) are available on GitHub

(https://github.com/vahidAK/NanoMethPhase/tree/master/scripts).”

2. Comment on the high incidence of polymorphic imprinting in the discussion.

We have revisited these results and also the new results from *ZFP57* motif analysis (See the next comment/point) and commented in the first paragraph of the Discussion section:

“DMRs that are detected in only a single study displayed higher variations across individuals compared to those detected by at least two studies. Therefore, lack of phasing at some novel DMRs in previous studies and higher variation in imprinted methylation at novel DMRs could explain the reason they were not detected previously. We also demonstrated that germline DMRs with a greater number of *ZFP57* motif tend to be more consistently imprinted across individuals suggesting motifs redundancy increases *ZFP57* recruitment and tolerance to any DNA sequence variation. However, due to the availability of DNA sequence in a limited number of samples, we were not able to examine sequence variation at the DMRs and the *ZFP57* binding motifs for any possible association with polymorphic imprinted methylation which will require further study.”

3. Test for sequence variation in ZFP57 binding sites in polymorphically methylated gDMRs if sequence information is available for the LCLs and other informative samples for which DNA methylation data exist.

We analysed DMRs for presence of the *ZFP57* binding motif (TGCCGC) and observed a significant correlation between number of binding motifs and number of individuals with partial methylation in their whole-genome bisulfite sequencing data at germline DMRs. However, as sequence variation information was only available for the 12 cell lines and very few DMRs had DNA sequence variation in these cell lines at a *ZFP57* motif, we were not able to derive any association between gDMR sequence variation and polymorphic imprinted methylation:

“Using *ZFP57* and *ZNF445* ChIP-seq peak calling information from human embryonic stem cells and the HEK 293T cell line (Imbeault et al., 2017; Takahashi et al., 2019), 44% of the novel gDMRs and 49% of the reported gDMRs were bound by *ZFP57* and/or *ZNF445* (Figure 3C; Supplementary File 4). Of these gDMRs, 89% had a *ZFP57* peak and 45% had a *ZNF445* peak. This highlights the importance of *ZFP57* as an important factor for the maintenance of imprinted methylation at gDMRs. 5′-TGC(5mC)GC-3′ is the canonical binding motif for *ZFP57* (Quenneville et al., 2011). 88% of the gDMRs with a *ZFP57* peak had at least one 5′-TGCCGC3′ motif, while 40% of the gDMRs without *ZFP57* peak had at least one 5′-TGCCGC-3′ motif in the human genome (GRCh38; Supplementary File 4). Moreover, at gDMRs the number of 5′TGCCGC-3′ motifs demonstrated a significant positive correlation with the number of individuals demonstrating partial methylation (Pearson = 0.54, p-value = 3.6e−07; Appendix 1 – figure 1). This suggests that a greater number of motifs provides more functional binding opportunities for *ZFP57* and also less likelihood that all *ZFP57* motifs could be perturbed through polymorphism or DNA sequence variation resulting in the imprinted methylation being less polymorphic.”

4. More rigor is required in the analysis of allelic enrichment of H3K36me3 and H3K27me3 at domains of extended PofO DNA methylation bias compared with those imprinted regions that do not, and compute the allelic read scores and provide an appropriate summary plot.

We have performed a binning-based binomial analysis of the allelic histone read counts to detect bins with significant allelic histone modifications at detected PofO methylated biased blocks and test blocks. We also included an appropriate plot to represent histone marks and the relation between H3K36me3 and H3K27me3 marks:

“To analyze allelic histone modifications and detect blocks of allelic histone marks at large blocks of PofO bias, we binned the genome into 10Kb intervals and performed a binomial test with Fisher’s combined p-value test to determine the significance of allelic read counts at 10Kb intervals with >3 informative heterozygous SNVs (Having at least 5 mapped reads) within each block in each sample. A 10Kb bin considered as significant for allelic histone mark if it had an adjusted P value <0.001 and if at least 70% of the SNVs within the 10Kb bin having ≥80% of the reads mapped to one allele. In total, 174 bins for H3K36me3 and 132 bins for H3K27me3 could be examined. Of these, 147 bins for H3K36me3 and 51 bins for H3K27me3 were significant. Thirty-eight bins were significant for both histone marks in the same sample. All the seven blocks demonstrated multiple significant bins for H3K36me3 at almost all the samples. *L3MBTL1*, *GPR1-AS/ZDBF2*, *GNAS/GNAS-AS1*, and *ZNF597/NAA60* demonstrated multiple significant H3K27me3 bins in majority of the samples and *KCNQ1OT1*, *PWS/AS*, and *ZNF331* had significant H3K27me3 bins at 3, 2, and 1 of the samples, respectively. H3K36me3 and H3K27me3 demonstrated mutual exclusive pattern and H3K36me3 appeared on the hypermethylated allele while H3K27me3 on the hypomethylated allele (Figure 8; Figure 8—figure supplement 1-6; Figure 9; Supplementary File 8).

To determine if allelic histone marks are unique to the PofO methylation-biased blocks, we examined allelic histone marks on several other imprinted clusters with strong ASE which did not display PofO bias methylation. For this, we examined *PPIEL*, *MEG3*, *MEST*, *DIRAS3*, *IGF2*, *MTRNR2L4*, and *ADNP2/PARD6G-AS1* clusters. Eighty-three bins for H3K36me3 and 138 bins for H3K27me3 could be examined at the seven test blocks. Of these, only 5 bins for H3K36me3 and 7 bins for H3K27me3 were significant and none of the bins were significant for both histone marks (Figure 9—figure supplement 1-8; Supplementary File 9). These results suggest that the blocks of PofO methylation bias in the gene body of active alleles are mediated by transcription and histone marks at their gene bodies.”

More details on these points are given below, as well as minor points that need to be addressed also.Reviewer #2 (Recommendations for the authors):The authors addressed some (not all) of the concerns raised in the first round of review, often differently than suggested (which is acceptable).The data availability however, is not acceptable in my view; given that the bioinformatic approach to DNA methylation phasing is computationally sophisticated and was published elsewhere already (Akbari, et al., 2021), it should be expected that the processed nanometh tracks (shown in Figure 3c, 6-8 and the extensive supplement) are made available via a data repository (e.g. GEO or Mendeley data). Inclusion of processed data tracks is obligatory for GEO submissions, and recreating these data tracks from the raw data places an undue burden on labs wishing to query these data for their loci of interest. These data tracks should include all chromosomes, including sex chromosomes, to enable others to re-run statistical analyses with their own (p-value and DNA methyation change) cutoffs.

We appreciate the reviewer’s positive evaluation of our effort to address the comments.

We have deposited methylation and histone modification tracks to Mendeley data repository (https://data.mendeley.com/datasets/f4k2gytbh5/1; doi: 10.17632/f4k2gytbh5.1).

Reviewer #3 (Recommendations for the authors):In my review of the original manuscript, I suggested that it was important that the study showed something new, either in relation to undiscovered imprinted loci with potential importance in human development, physiology or disease, or new concepts in imprinting regulation. Specifically, I recommended further analysis of:– polymorphic imprinting– analysis of ZFP57/ZNF445 binding sites within candidate gDMRs– stricter threshold for methylation levels in gametes to call candidate gDMRsand additional improvements including:– more focussed discussion– clear summary tableIn general, these points have been addressed well in the revised manuscript."Novel Imprinted DMRs Display Inter-Individual Variation"I think this is an important addition to the manuscript that demonstrates the majority of the novel DMRs identified in the current study could represent 'polymorphic' imprinting and they are not consistently methylated in blood samples from 87 individuals, in comparison to 'well-characterised' DMRs. Some of the novel DMRs exhibited partial methylation consistent with imprinted status in as few as 1-2% of individuals, which could explain why they had not been detected in previous studies. The relevant analysis is performed well.On the other hand, it is an omission that the authors do not comment on this high incidence of polymorphic imprinting in the discussion. The authors do need to return to this finding in their discussion. Although at this point, they are not able to provide much speculation for why these loci exhibit inter-individual variation, there are multiple implications of the finding.Regarding the opening line of this section, "Imprinted methylation can display variation across individuals due to environmental and genetic factors", I think this statement could be modified, as generally speaking imprinting (i.e., well-characterised imprints) is consistent between individuals and resistant to environmental factors (with procedures associated with assisted reproduction techniques possibly the most likely 'environment' to lead to instability of methylation at gDMRs). Therefore, rephrase to something like: "Although imprinted methylation is generally regarded is consistent between individuals and resistant to environmental factors, there are examples of polymorphic imprinting…….""Determination of Germline versus Somatic Status of Novel Imprinted DMRs"This section now includes the analysis of ZFP57/ZNF445 binding data – published ChIP-seq datasets from hESCs and HEK293 cells. Interestingly, 44% of the novel gDMRs have binding for one or both of these proteins in these cells, compared with 49% of characterised gDMRs. A difference in binding by these factors does not therefore appear to relate to the high incidence of polymorphic imprinting amongst novel DMRs, but the possibility that there is sequence variation in the ZFP57 binding sites between the sequenced samples has not been addressed.It is also welcomed that the authors apply the more stringent thresholds for gDMR classification in this section."Enriched Allelic H3K36me3 and H3K27me3 Histone Marks at Contiguous Blocks"It is interesting to see the differential allelic enrichment for these two histone modifications across the seven extended domains that display biased parent-of-origin DNA methylation. The authors draw a distinction with other imprinted domains without this extended PofO bias. This conclusion is based entirely on inspection of screenshots of ChIP-seq data, examples of which are provided in Figure 8 and Supplementary Figures 14-26. I think the authors could be a little more rigorous and compute the allelic read scores and provide an appropriate summary plot.

We appreciate reviewer’s positive evaluation of our revised manuscript.

1. It is an omission that the authors do not comment on this high incidence of polymorphic imprinting in the discussion. The authors do need to return to this finding in their discussion.

This comment has been addressed above under the “responses to the main points” point number 2.

2. Test for sequence variation in ZFP57 binding sites in polymorphically methylated gDMRs if sequence information is available for the LCLs and other informative samples for which DNA methylation data exist.

This comment has been addressed above under the “responses to the main points” point number 3.

3. The authors should be more rigorous in their analysis of allelic enrichment of H3K36me3 and H3K27me3 at domains of extended PofO DNA methylation bias compared with those imprinted regions that do not, and compute the allelic read scores and provide an appropriate summary plot.

This comment has been addressed above under the “responses to the main points” point number 4.

4. Rephrase statement: "Imprinted methylation can display variation across individuals due to environmental and genetic factors".

We have rephrased this sentence as suggested by reviewer:

“Although imprinted methylation is generally regarded as consistent between individuals and resistant to environmental factors, there are examples of polymorphic imprinting where imprinted methylation is not consistently observed across individuals.”